# A structure-based designed small molecule depletes hRpn13Pru and a select group of KEN box proteins

Xiuxiu Lu [1], Monika Chandravanshi[1], Venkata R. Sabbasani [2], Snehal Gaikwad[3], V. Keith Hughitt[3], Nana Gyabaah-Kessie[3], Bradley T. Scroggins[4], Sudipto Das[5], Wazo Myint [6], Michelle E. Clapp[7], Charles D. Schwieters[8], Marzena A. Dyba[9], Derek L. Bolhuis[10], Janusz W. Koscielniak[11], Thorkell Andresson[5], Michael J. Emanuele [12], Nicholas G. Brown [12], Hiroshi Matsuo [6], Raj Chari [7], Deborah E. Citrin[4], Beverly A. Mock [3], Rolf E. Swenson[2] & Kylie J. Walters [1]✉

Proteasome subunit hRpn13 is partially proteolyzed in certain cancer cell types to generate hRpn13Pru by degradation of its UCHL5/Uch37-binding DEUBAD domain and retention of an intact proteasome- and ubiquitin-binding Pru domain. By using structure-guided virtual screening, we identify an hRpn13 binder (**XL44**) and solve its structure ligated to hRpn13 Pru by integrated X-ray crystallography and NMR to reveal its targeting mechanism. Surprisingly, hRpn13Pru is depleted in myeloma cells following treatment with **XL44**. TMT-MS experiments reveal a select group of off-targets, including PCNA clamp-associated factor PCLAF and ribonucleoside-diphosphate reductase subunit M2 (RRM2), that are similarly depleted by **XL44** treatment. **XL44** induces hRpn13-dependent apoptosis and also restricts cell viability by a PCLAF-dependent mechanism. A KEN box, but not ubiquitination, is required for **XL44**-induced depletion of PCLAF. Here, we show that **XL44** induces ubiquitin-dependent loss of hRpn13Pru and ubiquitin-independent loss of select KEN box containing proteins.

The 26 S proteasome is responsible for regulated protein degradation in eukaryotes[1]. Its substrates are typically marked by post-translational modification with ubiquitin chains for recognition by ubiquitin receptors in the proteasome 19 S regulatory particle (RP)[2]. The RP caps either end of the proteasome 20 S core particle (CP), which is assembled from four heptameric rings with a hollow interior where substrate proteolysis occurs[3]. Small molecule inhibition of the CP is used to treat hematological cancers and an inhibitor of the CP variant immuno-proteasome (iP) present in hematopoietic cells or following pro-inflammatory signaling is used against autoimmune inflammatory myopathies. Additional therapeutic applications of the ubiquitin-proteasome pathway have emerged with proteolysis targeting chi-meras (PROTACs), bifunctional molecules that cause the degradation of a protein of interest by linking it to cellular machinery that catalyzes

ubiquitination[4]. PROTACs for androgen and estrogen receptor are in phase II clinical trials for prostate and breast cancer, respectively[5,6].

RP subunit hRpn13 is a substrate receptor with a Pru domain that binds to ubiquitin[7,8] with preference for K48-linked ubiquitin chains[9] due to interactions involving the ubiquitin linker region[10]. Adjacent and opposite to the ubiquitin-binding surface is a channel across the Pru domain where the intrinsically disordered extreme C-terminus of RP subunit hRpn2 binds[11–13]. hRpn13 also has a DEUBAD domain that binds and activates the deubiquitinase UCHL5/Uch37[14–16]. We recently found hRpn13 to interact with epigenetic factors histone deacetylase HDAC8 and arginine deiminase PADI4 in myeloma cells where it con-tributes to NF-κB processing and transcriptional regulation of cytos-keletal and other proteins[17]; hRpn13 interaction with HDAC8 was previously found in glioblastoma cells[18]. hRpn13 has emerged as a

therapeutic target, with a cysteine residue (Cys88) peripheral to the hRpn2-binding region[12,13] that is available for covalent modification by small molecules[19–21]. By linking one such molecule (**XL5**) to an E3 ligase-targeting compound to generate a PROTAC, a naturally existing hRpn13 fragment was discovered in myeloma cells that lacks the DEUBAD domain but has an intact Pru domain (hRpn13[Pru])[21]. hRpn13[Pru] is selectively ubiquitinated and degraded by hRpn13 PROTACs, which induce apoptosis in hRpn13[Pru]-producing cells[21]. The hRpn13 Pru and DEUBAD domains interact when free of binding partners[22] and the loss of DEUBAD removes any regulatory activities associated with the Pru:DEUBAD interaction as well as potential deubiquitination of hRpn13 by UCHL5/Uch37. An independent study further supported the therapeutic potential of hRpn13-targeting by a peptoid ligand (**KDT-11**) that restricts cell viability in myeloma cells[23].

In this work, we use a structure-based approach to identify a small molecule derivative of **XL5** (**XL44**) that depletes hRpn13[Pru] from myeloma cells. Unlike our previous hRpn13-targeting PROTACs, linking to an E3 ligase is not required for **XL44**-induced hRpn13[Pru] loss. By using TMT-MS experiments to examine target specificity, we find KEN box containing off-targets PCLAF (PCNA clamp associated factor) and RRM2 (ribonucleoside-diphosphate reductase subunit M2). PCLAF is a PCNA-binding co-factor that regulates DNA repair by facilitating PCNA interaction with DNA polymerase η at stalled replisomes[24] and drives cellular exit from quiescence to promote lung tumorigenesis by remodeling the DREAM (dimerization partner, retinoblastoma (RB)-like, E2F, and multi-vulval class B) complex[25]. RRM2 is one of two subunits of ribonucleotide reductase, which catalyzes formation of deoxyribonucleotide diphosphates from ribonucleoside diphosphates, a rate-limiting step in the production of deoxyribonucleotide triphosphates. RRM2 is transcriptionally upregulated at the G1/S phase transition[26,27] and at the end of the S phase, selected for ubiquitin-mediated degradation by the E3 ubiquitin ligase cyclin F[28]. Elevated RRM2 levels correlate with poor prognosis in patients with breast cancers[29] and decreased overall survival in patients with neuroblastoma[30] and epithelial ovarian cancer[31], with its knockdown reported to reduce cell proliferation in epithelial ovarian cancer cell lines[31]. The E3 ligase complex APC/C[FZR1] is reported to promote the degradation of PCLAF[32] and RRM2[33] by recognition of KEN boxes in their protein sequences. We provide evidence here for the dual targeting of hRpn13[Pru] with PCLAF and RRM2 through a mechanism for the latter proteins that requires their KEN box, a known degron for APC/C[FZR1] substrates[34].

## Results

### Structure-based screen for a potent hRpn13-binding compound

In an effort to develop a potent hRpn13-binding molecule, we created a small virtual library of 18 compounds chosen based on chemical similarity to **XL5** (Supplementary Table 1) and subjected these compounds to a covalent docking screen based on the hRpn13 Pru:**XL5** structure (PDB 7KXI, Fig. 1a) by using the Schrödinger software package. The covalent docking was designed for reversible Michael addition at hRpn13 Cys88, akin to **XL5** (Fig. 1a, left image). The top scoring compound (**XL44**) replaced the **XL5** benzoic acid group with an indoline ring (Supplementary Table 1) and a model structure suggested **XL44** and **XL5** form similar interactions with hRpn13 (Fig. 1a).

**XL44** binding to hRpn13 Pru was experimentally indicated by 2D NMR spectra (Fig. 1b), tryptophan quenching (Fig. 1c), and mass spectrometry (Fig. 1d). Addition of equimolar **XL44** caused signal shifting in $^1$H, $^{15}$N HSQC spectra recorded on 20 μM $^{15}$N-labeled hRpn13 Pru (Fig. 1b), including for the intended Michael receptor Cys88 (Fig. 1a, b). W108 is included among the amino acids shifted by **XL44** addition (Fig. 1b) and tryptophan quenching measured by differential scanning fluorimetry ($\lambda_{350}$) has been an effective indicator of hRpn13 Pru interaction[8,21]. We observed tryptophan quenching by serial dilution of **XL44** or **XL5** (for comparison) into 1 μM hRpn13 Pru, with greater

effects caused by **XL44** compared to **XL5** (Fig. 1c). Consistent with a covalent interaction, liquid chromatography-mass spectrometry (LC-MS) detected a product of the appropriate molecular weight for **XL44**-ligated hRpn13 Pru following hRpn13 Pru incubation with 10-fold molar excess **XL44** (Fig. 1d). An isomeric mixture of **XL44** was used for these biophysical experiments; however, separation of the two stereoisomers by silica gel column chromatography (Supplementary Fig. 1a) indicated no preference for the E or Z isomer, as assessed by 2D $^1$H, $^{15}$N HSQC experiments of 20 μM $^{15}$N-labeled hRpn13 Pru with 2-fold molar excess of E or Z **XL44** stereoisomer (Fig. 1b and Supplementary Fig. 1b).

Since **XL44** targets the proteasome-binding surface of hRpn13, we next tested whether hRpn2 prevents this interaction. The **XL44** indoline ring includes fluorine at the para position (Fig. 1e, left panel) that is observable by 1D $^{19}$F NMR. Experiments recorded on 50 μM **XL44** dissolved in NMR buffer (20 mM sodium phosphate, 50 mM NaCl, 10% DMSO-$d_6$, pH 6.5) without (grey) or with 2 mM DTT (black) indicates DTT-dependent signal shifting (Fig. 1e), indicating that **XL44** can react with DTT. These signals are lost and a broadened signal at −122.63 ppm is present following **XL44** incubation with equimolar hRpn13 Pru (Fig. 1e, orange). By contrast, no effect is observed following the incubation of equimolar hRpn2 (940-953) with **XL44** (Fig. 1e, red), indicating that **XL44** binds to hRpn13 but not hRpn2. Addition of 0.5 molar or equimolar equivalents of hRpn2 (940-953) to the pre-incubated mixture of **XL44** with equimolar hRpn13 Pru (orange) causes respective reduction (Fig. 1e, blue) or loss (Fig. 1e, navy) of hRpn13-bound **XL44** signal, with corresponding reappearance of the DTT and unbound **XL44** signals (Fig. 1e). This experiment indicates that **XL44** does not bind to hRpn2-bound hRpn13 Pru and that **XL44** cannot compete with hRpn2. hRpn13 is present both on and off proteasomes[12,15,17] and this data suggests that **XL44** targets extra-proteasomal hRpn13.

We next tested whether **XL44** can isolate hRpn13 from whole cell lysates of RPMI 8226 cells. We fused **XL44** to biotin, replacing the methoxy group with an amide linker (**XL44B**, Supplementary Fig. 2a, top panel). This modification reduced the affinity of the compound for hRpn13 Pru, as evaluated by 2D NMR, showing a reduction in **XL44**-bound signals (Supplementary Fig. 2b compared to Fig. 1b). Despite the weaker affinity, streptavidin-bound biotinylated **XL44** isolated hRpn13 from RPMI 8226 whole cell lysate (Fig. 1f and Supplementary Fig. 2c). For this experiment, biotin was used as a negative control and biotinylated **XL5** (**XL5B**, Supplementary Fig. 2a, bottom panel) as a positive control. Since hRpn13 is a ubiquitin receptor, we also immunoprobed for ubiquitin to find co-enriched ubiquitinated proteins (Fig. 1f). Co-enrichment of ubiquitinated proteins by hRpn13 binding prohibited our use of this approach to identify additional **XL44** targets. Altogether, these data indicate that **XL44** binds to hRpn13.

### hRpn13 Pru-XL44 structure by integrated NMR and crystallography

To further characterize the interaction of **XL44** with hRpn13 Pru, we used X-ray crystallography. Attempts to obtain crystals of hRpn13 Pru with **XL44** were only successful when ubiquitin was also included in the mixture, yielding a structure of hRpn13 Pru:ubiquitin:**XL44** solved to 2.1 Å resolution (Table 1). Each asymmetric unit of the crystal contained two molecules of the hRpn13 Pru:ubiquitin:**XL44** complex (Table 1). As expected, **XL44** was found to bind to a similar region as **XL5** (Fig. 1a and 2a), tethered at Cys88 by a covalent bond between this amino acid's sulfur atom and carbon 13 (C13) of **XL44** (Fig. 2a, insert). This surface is distinct from the hRpn13 ubiquitin-binding region, consistent with the pulldown of ubiquitinated proteins by **XL44B** (Fig. 1f).

The electron density for **XL44** was well resolved for the methoxybenzamide, central benzene and indolin-2-one rings, as was the thioester bond between C13 of **XL44** and hRpn13 Cys88 and the surrounding region, allowing stereo assignment of the two chiral centers at C12 and C13 to R and S respectively (Fig. 2a, insert). Analysis of **XL44**

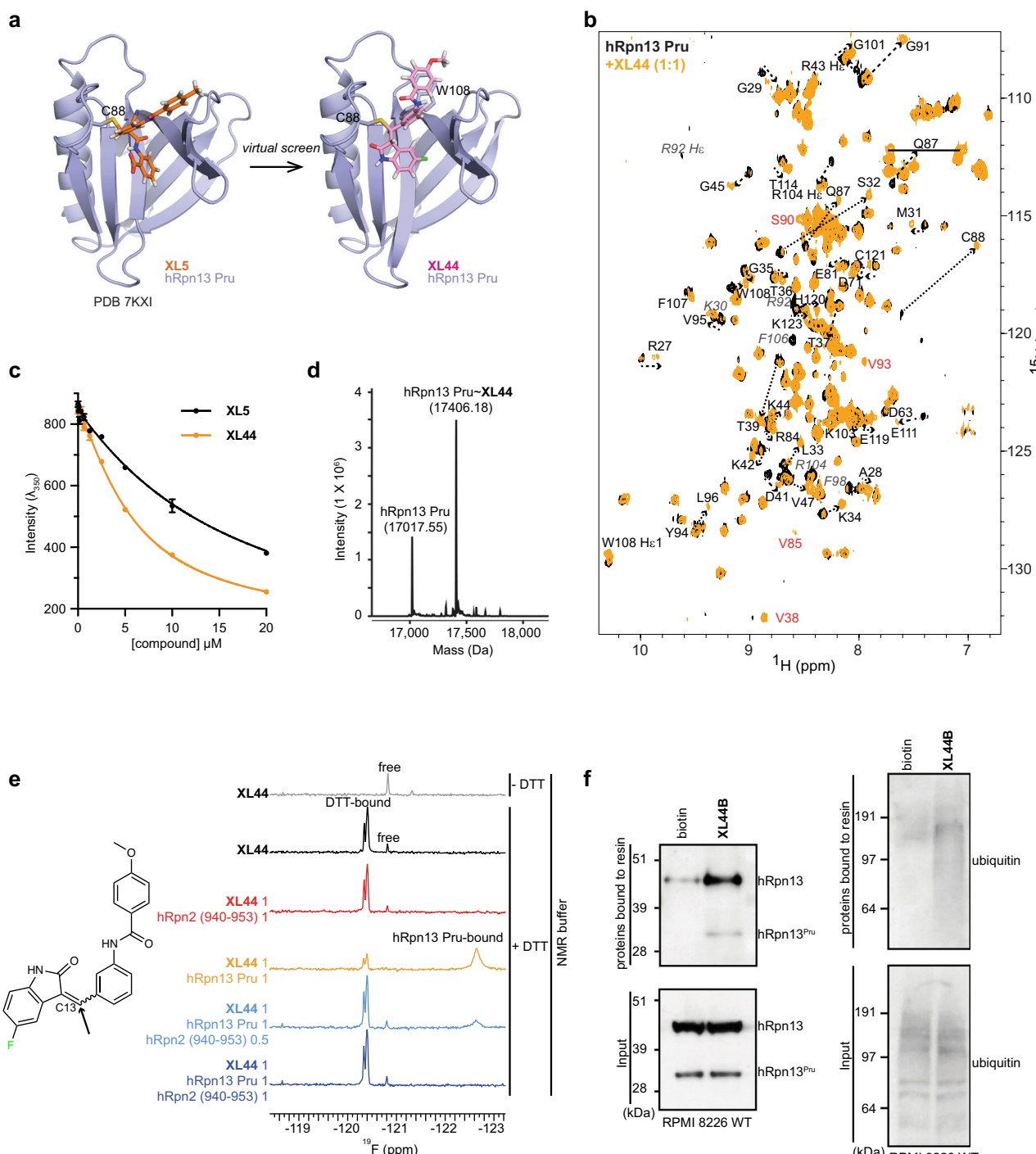

**Fig. 1 | Structure-based screen reveals hRpn13-binding compound XL44.**
**a** Ribbon diagram of the hRpn13 Pru-**XL5** structure (left, PDB 7KXI) used for virtual screening and the model structure of hRpn13 Pru with **XL44** (right) showing the compounds in stick representation. hRpn13 Pru, **XL5**, and **XL44** are colored purple, orange, and pink respectively with nitrogen, oxygen, hydrogen, and fluorine in blue, red, white, and green, respectively. **b** $^1$H, $^{15}$N HSQC spectra of 20 μM $^{15}$N-hRpn13 Pru (black) or with equimolar **XL44** (orange) in NMR buffer at 10 °C. Dashed arrows highlight the shifting of hRpn13 signals following **XL44** addition. Signals that disappear (italicized grey) or appear (red) are labeled. **c** Emission at 350 nm for 1 μM hRpn13 Pru following addition of **XL44** (orange) or **XL5** (black). The plots depict mean ± SD from three parallel recordings against compound concentration and were fit by the equation '[inhibitor] vs. response−Variable slope (four parameters)' in GraphPad Prism9. **d** LC-MS analysis of 2 μM hRpn13 Pru (MW:

17017.3 g/mol) incubated with 20 μM **XL44** for 2 h at 4 °C. The resulting **XL44** adduct and unmodified hRpn13 Pru are labeled with the detected molecular weight (Da) included. **e** **XL44** chemical structure (left) and $^{19}$F spectra (right) of **XL44** without (grey) or with (black) DTT, **XL44** with equimolar hRpn2 (940–953) (red) or hRpn13 Pru (orange), and 25 μM (light blue) or 50 μM (indigo) hRpn2 (940–953) with preincubated **XL44** and equimolar hRpn13 Pru. All spectra were recorded at 10 °C and 50 μM **XL44**. The reactive site at C13 is marked by an arrow (left). **f** Immunoblots with antibodies against hRpn13 (left panel) or ubiquitin (right panel) of RPMI 8226 lysates (bottom) or following a pulldown experiment with streptavidin-bound biotin or **XL44**B (top). For the pulldown, lysates from RPMI 8226 cells were incubated with streptavidin-bound by biotin or **XL44**B and washed to remove unbound proteins (top), this experiment was performed twice. Source data are provided as a Source Data file.

**Table 1 | Data collection and refinement statistics (molecular replacement)**

| | hRpn13: ubiquitin: XL44 complex |
|---|---|
| **Data collection** | |
| Space group | $P12_{1}1$ |
| Cell dimensions | |
| $a, b, c$ (Å) | 53.35, 59.28, 65.02, |
| $\alpha, \beta, \gamma$ (°) | 90, 113.23, 90 |
| Resolution (Å) | 42.66 – 2.1 (2.175 – 2.1) [a] |
| $R_{sym}$ or $R_{merge}$ | 0.06878 (0.4141) |
| $I / \sigma I$ | 9.01 (2.39) |
| Completeness (%) | 98.29 (96.92) |
| Redundancy | 3.4 (3.1) |
| **Refinement** | |
| Resolution (Å) | 2.10 |
| No. reflections | 32544 (3179) |
| $R_{work} / R_{free}$ | 0.2206/0.2466 |
| No. atoms | |
| Protein | 3018 |
| Ligand/ion | 58 |
| Water | 101 |
| *B*-factors | |
| Protein | 45.65 |
| Ligand/ion | 69.39 |
| Water | 42.85 |
| R.m.s. deviations | |
| Bond lengths (Å) | 0.283 |
| Bond angles (°) | 4.46 |

[a]Values in parentheses are for highest-resolution shell.

binding indicated that the **XL44** indolin-2-one ring interacts with hRpn13 Met31, Val85, and Phe106; the **XL44** central benzene ring interacts with hRpn13 Pro89; and the **XL44** methoxybenzamide ring interacts at opposite sides with hRpn13 Trp108 and Pro40 (Fig. 2b). Akin to the three aromatic rings of **XL44**, the electron density for these interacting residues is well resolved (Supplementary Fig. 3a, b). Except for the backbone atoms of Gly35 and Thr36, the hRpn13 β1 - β2 hairpin spanning Leu33 - Val38 was poorly defined (Fig. 2c). Moreover, the electron density for the methyl group of the **XL44** methoxybenzamide was also poorly defined (Fig. 2a, insert). Disorder in the β1 - β2 hairpin is intrinsic to the hRpn13 Pru fold, as NMR experiments probing backbone amide dynamics of apo hRpn13 Pru indicated greater high frequency motions in this region (Fig. 2d and Supplementary Fig. 3c).

To complement the crystal structure by providing information where the electron density map was poorly defined, we acquired NMR data on **XL44**-bound hRpn13 Pru. We recorded unambiguous intermolecular NOE interactions between hRpn13 Pru and **XL44** by acquiring a 3- or 2-dimensional $^{1}$H, $^{13}$C half-filtered NOESY experiment on samples of $^{13}$C-labeled hRpn13 Pru mixed with equimolar unlabeled **XL44** (Fig. 3a and Supplementary Fig. 3d–f) or of unlabeled hRpn13 Pru mixed with equimolar **XL44** with its central benzene ring $^{13}$C-labeled (**XL44**-$^{13}C_{6}$-CB) (Fig. 3b). These NOESY experiments are designed to reveal intermolecular interactions for atoms within 6 Å of each other[35]. Consistent with the saturation of the **XL44** alkene by Michael addition, a signal for a hydrogen atom (H10) attached to C13 (Supplementary Fig. 3d, left panel) was observed with intermolecular NOE interactions to hRpn13 Met31, Leu33, Val38, Val85, Cys88, and Val93 (Supplementary Fig. 3d, right panel).

In addition, interactions were recorded for the **XL44** methyl group (Fig. 3a) and hRpn13 Lys42 sidechain (Fig. 3a); these atoms were missing in the electron density map (Fig. 2a and Supplementary

Fig. 3b). The intermolecular interactions between the **XL44** methyl group and hRpn13 Met31, Pro40, Lys42, and Trp108 (Fig. 3a) were used as distance restraints for structure calculations in XPLOR-NIH to define the orientation of the **XL44** methyl group, freezing all other atoms to the **XL44**-bound hRpn13 crystal coordinates (Fig. 2a). Further optimization was done by manual fitting of NOE-based intermolecular interactions between **XL44** and hRpn13 Leu33, Val38 and Lys42 (Fig. 3a, b and Supplementary Fig. 3d–f) to reorient the side chains of these three residues in accordance with the experimental data. Finally, the structure was refined in CCP4 to obtain an NMR-assisted crystal structure that fit both the electron density map and intermolecular NOE information (Fig. 3c, d). Overall, the hRpn13 Pru:**XL44**:ubiquitin structure before and after NMR-based refinement are similar with a backbone r.m.s.d. of 0.117 Å for Cα atoms (Fig. 3c); however, the **XL44** methyl group and side chain atoms of hRpn13 Leu33, Val38 and Lys42 are reoriented to fit the NMR data. The refined structure has identical molecular interactions for the two molecules in the asymmetric unit (Supplementary Fig. 4a, b and Supplementary Table 2). Moreover, the binding interface between hRpn13 Pru and ubiquitin was unaltered compared to a previously determined crystal structure of hRpn13 Pru with ubiquitin and hRpn2 (PDB 5V1Y, Supplementary Fig. 4c)[13].

Whereas the crystal structure indicated RS stereochemistry for the C12 and C13 chiral centers respectively, the in silico structure predicted RR (C12,C13) stereochemistry (Fig. 4a). Only the RS stereoisomer of **XL44** was observed by crystallography, but the NMR data indicated heterogeneity at the C12 chiral center. For example, **XL44** H18 was recorded to have two slightly different chemical shift values (Supplementary Fig. 3d), with one signal from **XL44**-bound to hRpn13 with an S chiral center at C12, as indicated by the hRpn13 Val38 methyl groups exhibiting stronger NOE interactions to **XL44** H10 than **XL44** H18 (Supplementary Fig. 3d). By contrast, NOEs (Supplementary Fig. 3d) from **XL44** indoline-2-one H18 and H10 fit to either the R or S stereoisomer at C12. Specifically, NOEs to Leu33 (Fig. 4b) and Met31 (Fig. 4c) fit well to the R stereoisomer observed in the crystal structure, whereas those involving Val38, Val85, and Cys88 (Supplementary Fig. 3d) fit best to the S stereoisomer at C12, as modeled in Supplementary Fig. 4d. Altogether these data indicate that **XL44** binds to hRpn13 Pru with either R or S stereochemistry at C12 in solution even though only the R C12 chiral center crystallized. Neither the X-ray diffraction nor the NMR data provided evidence for the R chiral center at C13.

### XL44 extends along an hRpn13 Pru channel mimicking hRpn2 interactions

By contrast to **XL5**, which was predicted to bind to a different region than found experimentally[21], **XL44** bound where predicted (Fig. 4a, d), at the location of hRpn2 interaction[12,13] (Fig. 4e). The **XL44** methoxybenzamide group buried into an hRpn13 hydrophobic pocket formed by hRpn13 Pro40, Lys42, Val93 and Trp108, similar to hRpn2 Pro944 and Pro945 (Fig. 4d, e). By contrast, the **XL5** 4-methylbenzamide is directed towards hRpn13 Thr39 (Fig. 4f) where hRpn2 Pro947 is positioned (Fig. 4e) when hRpn2 binds hRpn13. This positioning causes the **XL5** 4-methylbenzamide group to be more solvent exposed compared to the **XL44** methoxybenzamide (Supplementary Fig. 4e).

Superposition of the hRpn13 Pru complexed structures shows translational movements of 4.2, 5.9, and 8.4 Å for the β1 - β2 hairpin upon binding **XL5, XL44** and hRpn2 relative to apo state, respectively (Supplementary Fig. 4f). The positioning of the hRpn13 β1 - β2 hairpin is more similar to the hRpn2-bound state for **XL44** compared to **XL5** and allows interaction with Trp108 as well as closer contacts with Leu33 and Phe106 (Fig. 4d–f). We found that the methoxybenzamide end of **XL44** could be extended and we therefore synthesized **XL52** in which the **XL44** methyl group is replaced with 5-(4-methylphenyl)-2H-tetrazole (Supplementary Fig. 5). Addition of **XL52** to $^{15}$N-labeled hRpn13 Pru caused signal shifting similar to **XL44** (Fig. 1b) indicating

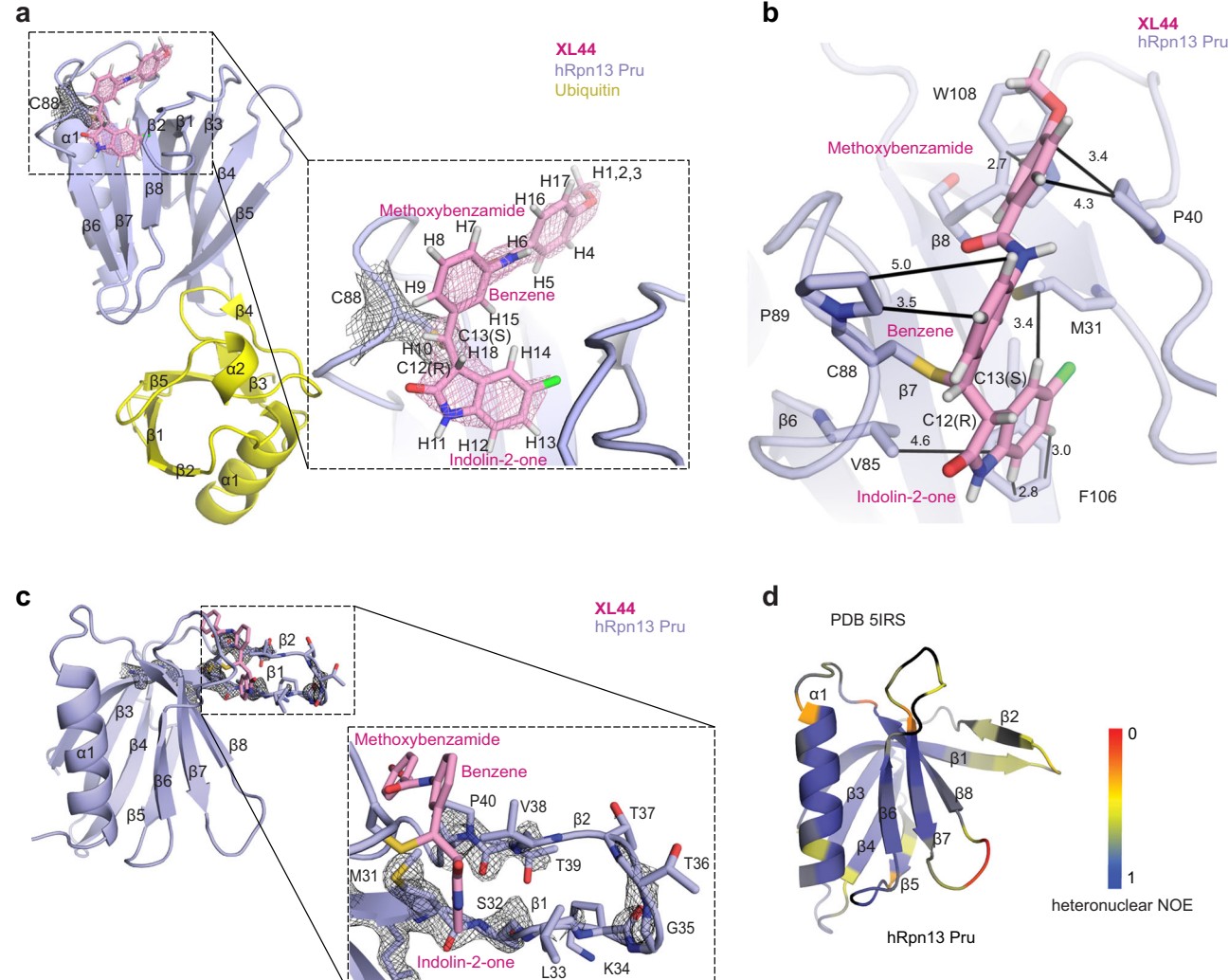

**Fig. 2 | Crystal structure of hRpn13 Pru bound to ubiquitin and XL44 with NMR data highlighting β1 – β2 hairpin dynamics. a** Structure of hRpn13 Pru (purple) bound to ubiquitin (yellow) with **XL44** (pink, stick rendering) covalently linked to hRpn13 Cys88. An expansion of the **XL44** region is included with the methoxybenzamide, benzene and indolin-2-one groups labeled (right). All protein secondary structural elements are labeled. The 2*Fo − Fc* electron density map is included for **XL44** (pink mesh) and Cys88 (black mesh) contoured at 1.7σ. Nitrogen, oxygen, hydrogen and fluorine atoms are colored blue, red, white and green respectively in all panels. The protons at the two stereocenters of **XL44** are labeled and adopt the R and S configuration for C12 and C13, respectively. **b** Expansion of the **XL44**-binding pocket of hRpn13 Pru with key interactions indicated by black lines and distances (Å). **c** 2*Fo − Fc* electron density map (black mesh) contoured at 1.7σ illustrating that except for the backbone of Gly35 and Thr36, the hRpn13 β1-β2 hairpin residues Leu33 - Val38 have poor density. **d** hetNOE analysis recorded on apo hRpn13 Pru mapped onto the structure of apo hRpn13 Pru (PDB 5IRS) with a scale bar. Source data are provided as a Source Data file.

that the methoxybenzamide extension is tolerated. As discussed below however this compound performed poorly in cellular assays.

The **XL44** central benzene ring forms hydrophobic interactions with hRpn13 Val38 at one side and Pro89 at the other (Fig. 4g). We tested the importance of this benzene packing by addition of a trifluoromethyl group (**XL53**) at the ortho position to find that this modification almost abolished binding to hRpn13 Pru (Supplementary Fig. 5), most likely due to steric clashes with the backbone oxygen of hRpn13 Gln87 (Fig. 4h). The **XL44** indolin-2-one ring is buried within a pocked formed by hRpn13 Leu33, Val38, Val85, Gln87, Cys88, Val93, Val95 and Phe106 (Fig. 4h). A weak hydrogen bond exists between the **XL44** indolin-2-one oxygen and the hRpn13 Gln87 side chain amide group (Fig. 4h). To experimentally evaluate the importance of the **XL44** indolin-2-one ring, we tested whether its replacement with 2-phenylthiazol-4-one (**XL54**) or 4H-1,4-benzothiazin-3-one (**XL55**) affects hRpn13 binding to find a reduction in both cases (Supplementary Fig. 5), most likely due to steric clashes with hRpn13 Phe106 or Leu33 (Fig. 4h). **XL44** H10 buries within a narrow pocket formed by

hRpn13 Gln87, Cys88, Leu33 and Val38 (Fig. 4h). We tested whether a methyl group could be accommodated at this site (**XL56**) and found reduced binding to hRpn13 (Supplementary Fig. 5), supporting the close contacts between H10 and hRpn13 Gln87 and Cys88 (Fig. 4h).

## XL44 induces loss of hRpn13^Pru and hRpn13-dependent apoptosis

To evaluate the biological effect of **XL44** on tumor cells, we tested whether it induces cell death by immunoblotting for the apoptosis markers cleaved caspase-9 or cleaved caspase-3. Lysates from RPMI 8226 cells treated for 24 h with 20 μM **XL44** or DMSO (vehicle control) were immunoprobed to find cleaved caspase-9 and cleaved caspase-3 prominent in **XL44**-treated cells whereas no effect was observed in the control experiment (Fig. 5a, top panel, lane 3 versus 1 and Fig. 5b, lane 2 versus 1). To test whether hRpn13 Pru is required for **XL44** induction of apoptosis, we used RPMI 8226 cells in which Exon 2 of the hRpn13 Pru domain is deleted and the resulting truncated hRpn13 protein (trRpn13) present at low abundance (trRpn13-MM2), as previously

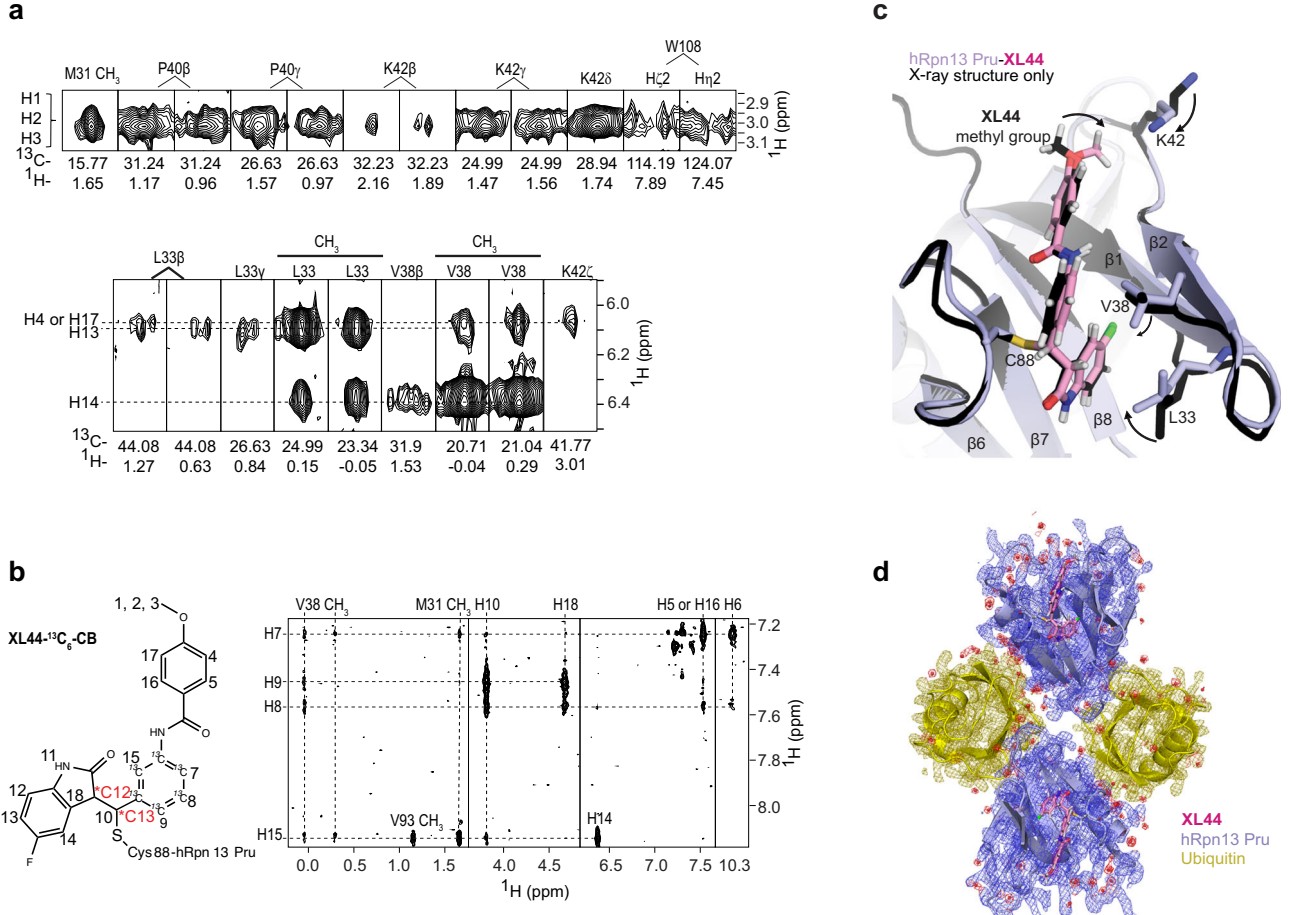

**Fig. 3 | XL44 interactions with hRpn13 Pru defined by integrated NMR and X-ray crystallography. a** Selected regions focusing on the **XL44** methyl group (top) or hRpn13 Leu33, Val38, and Lys42 (bottom) from a $^1$H, $^{13}$C half-filtered NOESY (100 ms) experiment acquired on a sample containing 0.4 mM $^{13}$C-labeled hRpn13 Pru and equimolar unlabeled **XL44** dissolved in NMR buffer. **b** Chemical structure of **XL44**-$^{13}$C$_6$-CB (left panel) and selected region from a $^1$H, $^1$H plane of a $^1$H, $^{13}$C half-filtered NOESY (100 ms) experiment acquired without incrementing the $^{13}$C dimension on a sample with 0.4 mM hRpn13 Pru mixed with equimolar of

**XL44**-$^{13}$C$_6$-BA in NMR buffer (right panel). **c** Overlay of X-ray structure (black) and after refinement by integrating the NMR data (purple) for the **XL44**-bound region. The methyl group of **XL44** as well as side chains of Leu33, Val38, and Lys42 are reoriented in the integrated structure (shown as black arrows) by NOE data. **d** $2Fo − Fc$ electron density map contoured at 1.7σ displayed for one asymmetric unit of the ternary hRpn13 (purple), ubiquitin (yellow) and **XL44** (pink) complex. Nitrogen (blue), oxygen (red), hydrogen (white), and fluorine (green) atoms of **XL44** and bound water molecules (red spheres) are displayed.

described[17,21]. Apoptosis was not induced in trRpn13-MM2 cells treated with **XL44** (Fig. 5a, top panel, lane 4 versus 2 and Fig. 5b, lane 4 versus 3). We next tested and found that **XL44**-induced apoptosis can be rescued by reintroduction of hRpn13 into trRpn13-MM2 cells by using a lentiviral cDNA-based approach. Immunoblotting indicated successful reintroduction of hRpn13 into trRpn13-MM2 cells and its proteolysis to hRpn13$^{Pru}$ (Fig. 5b, lower panels, lanes 5-6). **XL44** induced the production of cleaved caspase-3 in these hRpn13-expressing trRpn13-MM2 cells (Fig. 5b, lane 6 versus 5). Altogether, these data indicate that hRpn13 is required and sufficient for **XL44**-induced apoptosis (Fig. 5a, b).

RPMI 8226 cells produce an hRpn13 fragment (hRpn13$^{Pru}$) that contains an intact hRpn2-binding Pru domain and interdomain linker region but no UCHL5-binding DEUBAD domain (Fig. 5a, bottom panel). We previously found hRpn13$^{Pru}$ to be degraded by **XL5**-derived PRO-TACs (**XL5-VHL-2**) but not **XL5** itself[21]. Surprisingly, we found **XL44** treatment to cause a reduction in hRpn13$^{Pru}$ levels in RPMI 8226 WT cells (Fig. 5a, lane 3 versus lane 1 and Fig. 5b, lane 2 versus lane 1) and in hRpn13-expressing trRpn13-MM2 cells (Fig. 5b, lane 6 versus lane 5). We determined a half-maximal depletion concentration (D$_{ep}$C$_{50}$) value with corresponding concentration-dependent increase in cleaved caspase-9 (Fig. 5c) of 19 μM, reduced compared to the 39 μM of DC$_{50}$

of the **XL5**-derived PROTAC **XL5-VHL-2**[21]. Moreover, following treatment of RPMI 8226 cells with 20 μM **XL44**, a d$_{1/2}$ value (differential protein abundance following treatment) of 12 h was measured (Fig. 5d), shorter than the 16 h d$_{1/2}$ of **XL5-VHL-2**[21].

Since **XL52** also bound hRpn13 Pru, we tested its effect on RPMI 8226 cells but found that it did not restrict cell viability (Supplementary Fig. 6a). We hypothesized that this lack of activity might be due to its inability to permeate the cells. An artificial membrane permeability assay (Eurofin) validated this hypothesis (Supplementary Table 3). Therefore, we focused our further analyses on **XL44**.

## XL44 restricts the viability of multiple cancer cell lines

We next evaluated cell viability in RPMI 8226 WT and trRpn13-MM2 cells following 48-hour treatment with **XL44** by measuring metabolism with an MTT (3-(4,5-dimethylthiazol-2-yl)-2,5-diphenyltetrazolium bromide) assay. This experiment revealed reduced viability in both cell lines, with IC$_{50}$ values of 6.2 ± 0.5 μM and 4.5 ± 0.2 for WT and trRpn13-MM2 cells, respectively (Fig. 5e), indicating a restriction of viability that is hRpn13-independent in trRpn13-MM2 cells. Each **XL44** stereoisomer restricted RPMI 8226 cells with similar efficacy (Supplementary Fig. 6b) consistent with the aforementioned NMR data (Fig. 1b and Supplementary Fig. 1b).

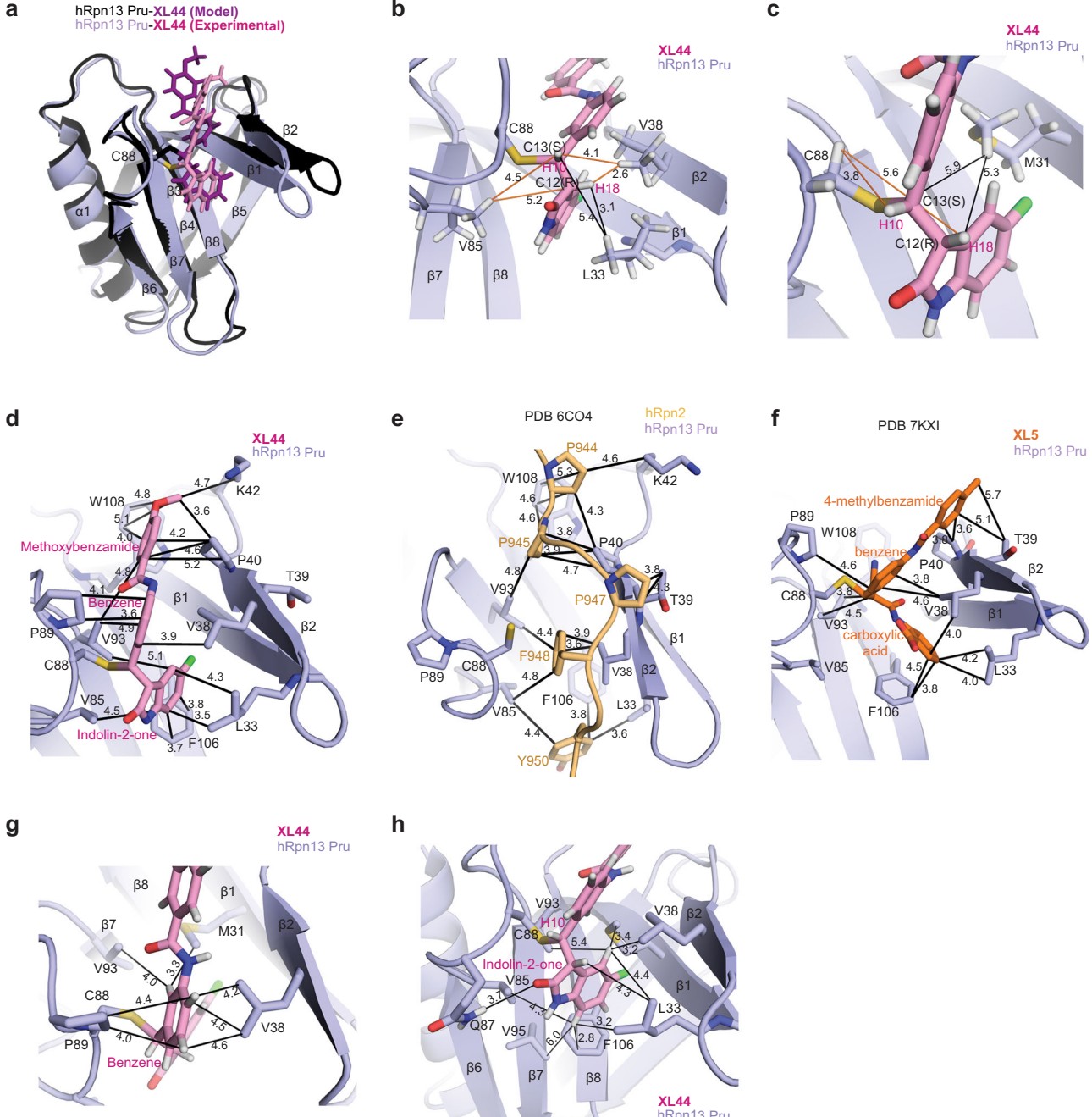

**Fig. 4 | XL44 forms extensive hydrophobic interactions with hRpn13 Pru.**
**a** Overlay of the model (**XL44**, dark pink; hRpn13 Pru, black) and experimental (**XL44**, light pink; hRpn13 Pru, purple) structures with ubiquitin omitted. **b**, **c** Expanded view of **XL44**-bound hRpn13 showing close contacts between **XL44** H10 and H18 and hRpn13 Leu33, Val38, Val85 **b** and Met31 **c** with distances (Å) included. Interactions consistent with detected NOEs are represented with black lines and those that do not support an R stereo-configuration at C12 are displayed with orange lines. **d**–**f** Comparative structural analysis of hRpn13 Pru bound with **XL44** (pink, **d**), hRpn2 peptide (940–953, **e**) (light orange) or **XL5** (orange, **f**) with similar interactions highlighted (black lines with distances (Å) included) for the **XL44** methoxybenzamide and hRpn2 Pro944 and Pro945. Distinct interactions are highlighted for **XL5** 4-methylbenzamide. **g**, **h** Expanded view of extensive hydrophobic interactions between the **XL44** central benzene **g** and indolin-2-one **h** rings with hRpn13 Pru residues. Colored as in Fig. 2a.

**XL44** also restricted the viability of other cell types, including breast cancer cell line MCF7, multidrug-resistant ovarian cancer cell line OVCAR-4, and metastatic carcinoma model cell line OVCAR-5, with $IC_{50}$ values of $2.3 \pm 0.5\,\mu M$, $3.9 \pm 0.2\,\mu M$ and $7.1 \pm 0.9$, respectively (Fig. 5f). By contrast, no effect was observed in non-cancerous human fibroblast 1634 and HS5 bone marrow-derived stem cells (BMSCs) up to 15 μM, albeit 25 μM and 50 μM **XL44** reduced the viability of HS5 cells (Fig. 5g). **XL44**-induced cell death was also measured in WT or trRpn13-MM2 cells by lactate dehydrogenase production following 48-

hour treatment to find an equivalent cell killing effect (Fig. 5h). These data indicate that **XL44** restricts the viability of multiple cancer cell lines.

## XL44 causes loss of PCLAF, its downstream target PTTG1, and RRM2

We next performed tandem mass tag mass spectrometry (TMT-MS) on lysates from RPMI 8226 WT cells treated for 8 h with 20 μM **XL44** or DMSO (vehicle control) in triplicate. 6469 proteins were detected

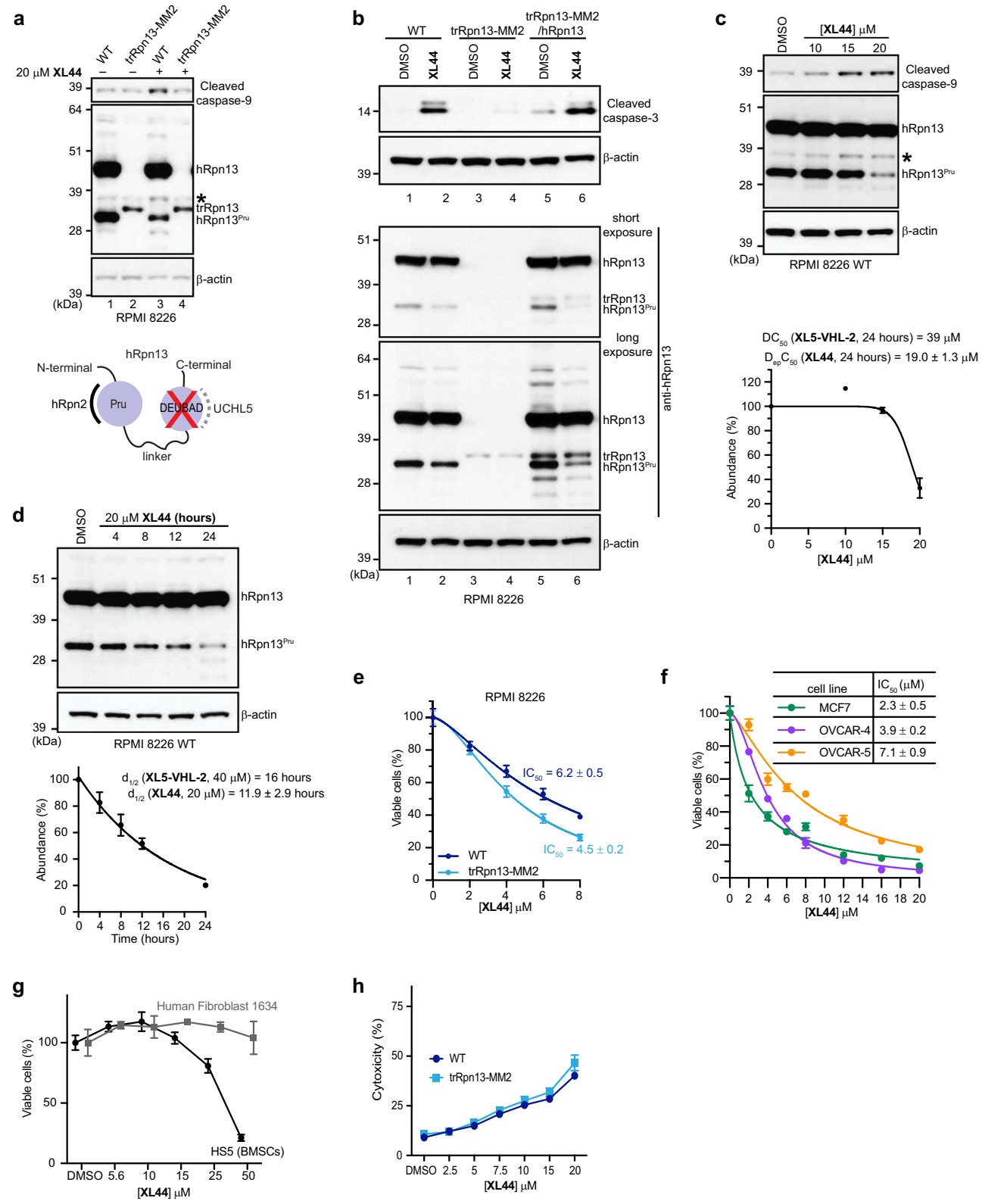

by this analysis, with five proteins identified to be ≥ 0.5-fold reduced and with p-values < 0.001; these include PCLAF, RRM2, proline rich 11 (PRR11), ribosomal biogenesis factor (RBIS), and cell division cycle 25 A (CDC25A) (Fig. 6a). To interrogate these findings further, we immunoprobed lysates from RPMI 8226 WT or trRpn13-MM2 cells following 24-hour treatment with 20 μM XL44 or DMSO (vehicle control) for proteins with available antibodies, including PCLAF and RRM2. PCLAF was depleted in RPMI 8226 WT and trRpn13-MM2 cells

following XL44 treatment and was also at reduced levels in trRpn13-MM2 compared to WT cells (Fig. 6b). These results were corroborated in an experiment with RPMI 8226 WT cells treated with varying concentrations of XL44 (Fig. 6c). Pituitary tumor-transforming gene 1 (PTTG1) is a downstream target of PCLAF[36] and PRR11[37] and PTTG1 was also reduced by XL44 treatment (Fig. 6b, c) and similar to PCLAF, reduced in trRpn13-MM2 (Fig. 6b). We attempted to evaluate PRR11 and CDC25A protein levels, but the commercial antibodies tested

**Fig. 5 | XL44 induces apoptosis in an hRpn13-dependent manner and reduces hRpn13^Pru levels.** a–c Lysates from RPMI 8226 WT **a**–**c**, trRpn13-MM2 **a**, **b** or trRpn13-MM2 with reintroduced hRpn13 (**b**, trRpn13-MM2/hRpn13) treated for 24 h with 20 µM (a and b) or indicated concentration **c** of **XL44** or DMSO (control) were immunoprobed as indicated. *, cleaved caspase-9. Cartoon (**a**, bottom) displaying the hRpn13 domains (purple circles), including hRpn2-binding Pru and UCHL5-binding DEUBAD, with the intrinsically disordered region as a black line. A red 'X' denotes the deletion of DEUBAD from hRpn13^Pru. **d** Immunoblots as indicated of lysates from RPMI 8226 cells treated for indicated times with 20 µM **XL44** or DMSO (0 h, vehicle control). Abundance (mean ± SD) plotted against **XL44** concentration (**c**, µM) or time (**d**, hours) derived as a percentage by intensity ratio of hRpn13^Pru normalized to β-actin ($I_{hRpn13^{Pru}}/I_{\beta\text{-}actin}$)sample divided by DMSO-treated values and multiplied by 100 for two replicates. Data fit by '[Inhibitor] vs. normalized response - Variable slope' (**c**) or 'One phase decay' (**d**) in GraphPad Prism9. $D_{ep}C_{50}$ (**c**) and $d_{1/2}$

(**d**) values are included, as are values for **XL5-VHL-2**[21]. **e** MTT assay for RPMI 8226 WT (blue) or trRpn13-MM2 (light blue) cells treated for 48 h with indicated **XL44** concentration. Data represent mean ± SD of $n = 6$ biological replicates. Viability calculated as ($\lambda_{570}$)sample/($\lambda_{570}$)control*100 (%). **f** As in **e**, but with MCF7 (green), OVCAR-4 (purple) or OVCAR-5 (orange) cells. Data represent mean ± SD of $n = 3$ biological replicates. **g** MTS assay of human fibroblast 1634 (grey) or HS5 (BMSCs) (black) cells treated for 48 h with indicated **XL44** concentrations. Data represent mean ± SD of $n = 4$ biological replicates. IC$_{50}$ values in **e**, **f** analyzed by '[Inhibitor] vs. normalized response - Variable slope' in GraphPad Prism9. **h**, LDH activity measured by CQUANT LDH Cytotoxicity Assay Kit for RPMI 8226 WT (blue) or trRpn13-MM2 (light blue) cells treated for 24 h with indicated **XL44** concentration. Data represent mean ± SD of $n = 4$ biological replicates. Cytotoxicity calculated as ((LDH activity)sample-(LDH activity)spontaneous))/((LDH activity)Maximum-(LDH activity)spontaneous))*100 (%). Source data are provided as a Source Data file.

were not specific, and we were unable to find validated commercial antibodies against RBIS.

Two bands were detected for RRM2 (Fig. 6b, c) by mouse anti-RRM2 antibodies, with the upper band reduced in **XL44**-treated RPMI 8226 WT or trRpn13 cells (Fig. 6b) with dose-dependency (Fig. 6c). HCT116 cells are more easily transfected than RPMI 8226 cells and transfection with 50 nM RRM2 or scrambled (as a control) siRNAs caused the upper band and not the lower band to be reduced (Supplementary Fig. 7), indicating the lower band to be non-specific. Only the upper band was detected by rabbit anti-RRM2 antibodies (Fig. 6b), which also indicated **XL44**-induced loss of RRM2 in WT and trRpn13-MM2 cells.

Treatment of RPMI 8226 cells with varying **XL44** concentration indicated a $D_{ep}C_{50}$ value of 8.4 ± 1.3 or 7.8 ± 1.7 µM for PCLAF or RRM2 (Fig. 6d), respectively. Moreover, $d_{1/2}$ values for PCLAF and RRM2 following 20 µM **XL44** treatment were measured to be 3.4 ± 1.3 and 11.2 ± 5.8 h, respectively (Fig. 6e, f).

Motivated by these results, we evaluated the association of PCLAF, PTTG1 and RRM2 expression levels with overall survival in a large-scale myeloma patient dataset[38]. These analyses indicated strong correlation, with p-values of <0.0001 for each (Fig. 6g and Supplementary Fig. 8a). As expected, PCLAF and PTTG1 expression was also correlated (Supplementary Fig. 8b). A weaker correlation was observed for the hRpn13-expressing gene *ADRM1*, with a p-value of 0.07 (Supplementary Fig. 8c). We note however that these data are for full length hRpn13 and perhaps a stronger correlation exists for hRpn13^Pru abundance, which is not available in this dataset.

Altogether, these data indicate that in addition to inducing the loss of hRpn13^Pru, **XL44** causes PCLAF, its downstream target PTTG1, and RRM2 to be reduced in cells. However, **XL44**-induced reduction of PCLAF, PTTG1 and RRM2 in trRpn13-MM2 cells did not induce apoptosis, as assessed by cleaved caspase-9 (Fig. 6b), providing further evidence that **XL44**-induced apoptosis is hRpn13-dependent.

To dissect further the mechanism of RRM2 and PCLAF reduction by **XL44**, we tested whether these two proteins can be isolated from RPMI 8226 lysates by **XL44B**-bound streptavidin, as was done for hRpn13 (Fig. 1f). RRM2 but not PCLAF was enriched by streptavidin-bound **XL44B** and migrated at a molecular weight characteristic of the unmodified (45 kDa, not ubiquitinated) protein (Fig. 6h). We next tested for PCLAF interaction with **XL44** by NMR. Comparison of 2D $^1$H, $^{15}$N HSQC spectra recorded on 20 µM $^{15}$N-labeled PCLAF without and with 10-fold molar excess **XL44** yielded no spectral changes (Supplementary Fig. 9a), indicating that **XL44** does not bind to PCLAF in this assay. We also tested for their interaction by LC-MS, but incubation of **XL44** with PCLAF yielded <1% **XL44**-adducted PCLAF (Supplementary Fig. 9b). Altogether, these data suggest that **XL44** can interact directly with RRM2, but not PCLAF.

### hRpn13^Pru is generated spontaneously by proteasomes
We next sought to test the role of the proteasome and ubiquitination machinery in **XL44**-induced loss of hRpn13^Pru, PCLAF and RRM2. We

previously found that the presence of hRpn13^Pru in cells requires proteasome activity[21]. To test whether the proteasome can spontaneously generate hRpn13^Pru by partially proteolyzing hRpn13, we incubated full-length hRpn13 produced from *E. coli* with commercially available 26 S proteasomes for 10 min and immunoprobed the mixture with anti-hRpn13 antibodies. This experiment revealed lower molecular weight hRpn13 bands following exposure to proteasomes, with an hRpn13 proteolyzed product that is consistent with the molecular weight of hRpn13^Pru (Fig. 7a).

To further study hRpn13 proteolysis in cells, we used proteasome inhibitor carfilzomib, ubiquitin E1-activating enzyme (UAE) inhibitor **MLN7243** (UAEi), and Nedd-8 activating enzyme (NAE) inhibitor **MLN4924** (NAEi) to assess the role of the proteasome, UAE, and NAE enzymes, respectively. Treatment with 100 nM carfilzomib for 24 h led to reduced levels of hRpn13^Pru (Fig. 7b, lane 3 versus lane 1), in support of proteasome activity being required to generate this hRpn13 cleavage product. Carfilzomib treatment resulted in higher molecular weight bands being detected between 51 and 64 kDa for hRpn13, which were also present to a lesser extent in **XL44**-treated cells (Fig. 7b, lane 2). Based on our previous experiments with **XL5**-based PROTACs[21], we hypothesized these hRpn13 species to be ubiquitinated hRpn13 and as expected, their reduction was observed following **XL44** co-treatment with 1 µM UAEi for 22 h (with 2 h of UAEi pre-treatment), with no impact by co-treatment with NAE**i** (Fig. 7b, lane 6). Correspondingly, lower molecular weight hRpn13 species were increased with UAE inhibition (Fig. 7b, lane 5). Depletion of hRpn13^Pru levels by **XL44** was not affected by UAEi (Fig. 7b, lane 5, and Supplementary Fig. 10a), indicating ubiquitin-independent proteolysis of hRpn13^Pru to the smaller hRpn13 species (Fig. 7c). Collectively, these findings indicate a stepwise process of spontaneous proteolysis of hRpn13 by the proteasome to generate hRpn13^Pru and **XL44**-induced further proteolysis to fragments which are cleared in a ubiquitin-dependent manner that does not involve neddylation or cullins (Fig. 7b, c).

### XL44-induced depletion of PCLAF and RRM2 requires proteasomes
The levels of PCLAF were also immunoprobed in the experiment of Fig. 7b to find an increase in carfilzomib-treated RPMI 8226 cells (Fig. 7b, lane 3 versus lane 1). Moreover, 22-hour co-treatment of 100 nM carfilzomib (with 2 h pre-treatment) with 20 µM **XL44** led to increased PCLAF abundance compared to untreated cells (Fig. 7b, lane 4 versus 1) whereas negligible effects were observed by co-treatment of **XL44** with UAEi (Fig. 7b, lanes 5 compared to 2 and Supplementary Fig. 10a). These findings suggests that PCLAF is turned over by proteasomes with and without **XL44** treatment by a ubiquitin-independent process (Fig. 7d), consistent with a previous report of it being intrinsically disordered[39].

**XL44** as a single treatment induced loss of RRM2 (Fig. 7e, lane 2 compared to 1), but its co-treatment with carfilzomib (Fig. 7e, lane 4)

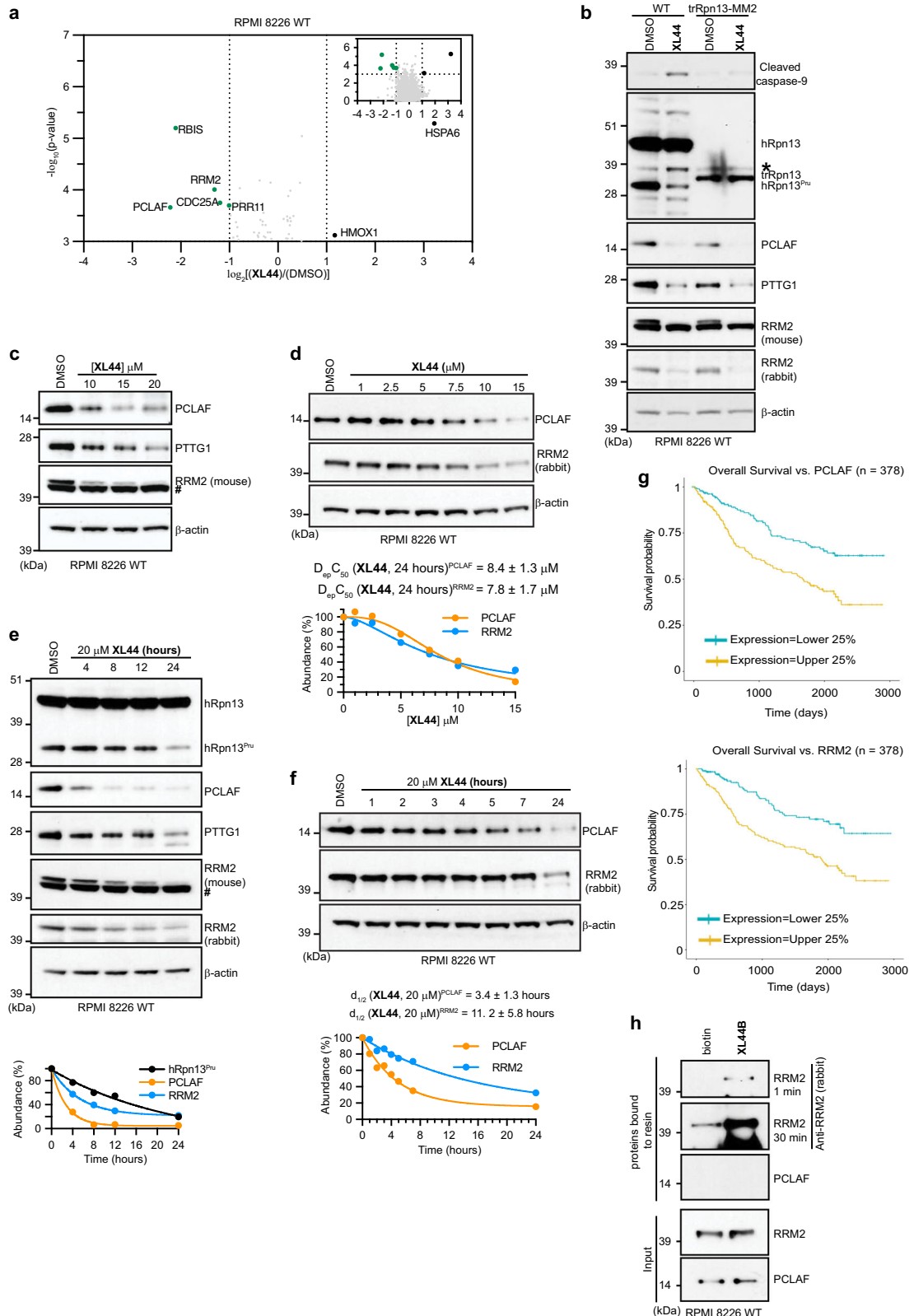

and to a lesser extent UAEi (Fig. 7e, lane 6) prevented this reduction. Carfilzomib as a single treatment or co-treatment with **XL44** and UAEi co-treatment with **XL44** caused a lower molecular weight band to appear suggesting partial proteolysis of RRM2; no such effect was observed with NAEi (Fig. 7e, lane 7). Altogether, these data indicate that **XL44**-induced depletion of PCLAF and RRM2 requires proteasome activity (Fig. 7d).

## A KEN box is required for XL44-induced depletion of PCLAF

The E3 ligase complex APC/C^FZR1 is reported to promote the degradation of PCLAF[32] and RRM2[33] by recognition of KEN boxes in their protein sequences. We tested whether **XL44** enhances ubiquitination of PCLAF by APC/C^FZR1. 5 μM fluorescently labeled PCLAF was pre-incubated with equimolar FZR1 and 8-fold molar excess **XL44** or DMSO (as a control) for 1 h, followed by addition of ubiquitin and other APC/C

**Fig. 6 | XL44 reduces protein levels of PCLAF, its downstream target PTTG1 and RRM2. a** Volcano plot from quantitative TMT proteomics analysis performed in triplicate for RPMI 8226 cell lysates treated for 8 h with 20 μM **XL44** or DMSO (control). *p*-values calculated by two-tailed two-sample equal variance t-test. Dashed lines indicate -log$_{10}$0.001 (horizontal), log$_2$0.5 (vertical) and log$_2$2 (vertical). Proteins reduced at ≥0.5-fold with *p*-values < 0.001 indicated by green dots. **b–c** Repeated experiments of Fig. 5a or Fig. 5c immunoblotting for cleaved caspase-9 **b**, hRpn13 **b**, PCLAF, PTTG1, or RRM2 with mouse **b**, **c** or rabbit **b** primary and secondary antibodies. **d** Lysates from RPMI 8226 cells treated with indicated **XL44** concentration or DMSO (control) for 24 h immunoprobed for PCLAF, RRM2 (rabbit) and β-actin. **e** Repeated experiments of Fig. 5d with inclusion of immunoblots for PCLAF, PTTG1 and RRM2 (mouse and rabbit). **f** Immunoblots of RPMI 8226 cell lysates treated for indicated times with 20 μM **XL44** or DMSO (0 h, control) detecting PCLAF, RRM2 or β-actin. Percentage (%) in **d–f** calculated as the ratio of intensities for PCLAF **d–f**, RRM2 **d–f**, or hRpn13$^{Pru}$ **e** normalized to β-actin

($I_{PCLAF/RRM2/hRpn13}$$^{Pru}$/$I_{β-actin}$)$_{sample}$, divided by DMSO control and multiplied by 100. Percentage (%) of PCLAF (orange), RRM2 (light blue) **d–f** or hRpn13$^{Pru}$ (black) **e** plotted against **XL44** concentration (**d**, μM) or time (**e**, **f**, hours) and fit by '[Inhibitor] vs. normalized response - Variable slope' **d** and 'One phase decay' **e–f** in GraphPad Prism9. D$_{ep}$C$_{50}$, **d** and d$_{1/2}$ **f** values are included. "#" in **b**, **c** and **e** indicates a non-specific band detected by anti-RRM2 mouse antibodies. **g** Graphical display of the correlation (*p*-value < 0.0001) between high (yellow) and low (blue) PCLAF (top panel) or RRM2 (bottom panel) expression levels with overall survival for patients with myeloma. **h** Immunoblots with antibodies against RRM2 (1 or 30 min) or PCLAF of streptavidin-bound **XL44**B or biotin (negative control) following incubation with RPMI 8226 cell lysates and subsequent wash (top panels) and corresponding input are included (lower panels). Experiments in 6b, 6c, 6e and 6 h were performed twice; 6d and 6 f were performed once. Source data are provided as a Source Data file.

subunits to initiate ubiquitination of PCLAF. **XL44** did not affect PCLAF ubiquitination by APC/C$^{FZR1}$ (Supplementary Fig. 10b), consistent with the ubiquitin-independent clearance of PCLAF by **XL44** (Fig. 7b and Supplementary Fig. 10a).

To test whether the KEN box sequence is required for **XL44**-induced depletion of PCLAF, we transfected HCT116 cells with 1 μg empty vector (pcDNA3.1 + N-HA), HA-PCLAF (WT) or HA-PCLAF with alanine substitutions at KEN box amino acids K78 and E79 (K78A and E79A) and after 48 h, treated the cells with DMSO (control) or 20 μM **XL44**. **XL44** treatment caused reduction of endogenous PCLAF (Fig. 7f, lane 2 versus lane 1) and overexpressed HA-PCLAF (Fig. 7f, lane 4 versus lane 3), but not of HA-PCLAF (KE-AA) (Fig. 7f, lane 6 versus lane 5). Altogether, these data indicate that **XL44** depletion of PCLAF occurs by a mechanism involving its KEN box, without altering its ubiquitination by APC/C$^{FZR1}$.

## XL44 restricts cell viability by dually targeting hRpn13$^{Pru}$ and PCLAF

To test whether **XL44** restriction of cell viability requires PCLAF, we used HCT116 WT and trRpn13 cells; like trRpn13-MM2 cells, trRpn13 HCT116 cells produce a truncated form of hRpn13 that begins at M109, thereby deleting much of the Pru domain[12]. We knocked down PCLAF levels by siRNA in HCT116 WT (Supplementary Fig. 10c, lane 2 versus lane 1) and trRpn13 (Supplementary Fig. 10c, lane 6 versus lane 5) cells (with comparative scramble siRNA) and treated with **XL44**. PCLAF was reduced by **XL44** in WT (Supplementary Fig. 10c, lane 3 versus 1) and trRpn13 (Supplementary Fig. 10c, lane 7 versus lane 5) cells, with greatest loss by the co-treatment of **XL44** with PCLAF siRNA (Supplementary Fig. 10c, lane 4 versus lane 2 and lane 8 versus lane 6).

In parallel with the experiment of Supplementary Fig. 10c, we measured cell metabolism by the MTT assay of Fig. 5e. **XL44** restriction of cell viability was reduced in HCT116 trRpn13 compared to WT cells (Fig. 7g, and Supplementary Fig. 11, left panel), indicating greater potency when hRpn13 Pru domain is present. Moreover, knockdown of PCLAF reduced **XL44** restriction of cell viability in WT cells compared to the scramble control (Fig. 7g, and Supplementary Fig. 11, second panel) and to a lesser extent, in trRpn13 (Fig. 7g, and Supplementary Fig. 11, third panel) cells. **XL44** was less potent in HCT116 trRpn13 cells treated with PCLAF siRNA compared to the single loss of either hRpn13 or PCLAF, with a clear difference compared to WT control cells (Fig. 7g and Supplementary Fig. 11, right panel). Therefore, we concluded that PCLAF contributes to **XL44** restriction of cell viability.

## Discussion

In an effort to design a stronger and more potent hRpn13$^{Pru}$ binder, we unexpectedly discovered a small molecule that depletes hRpn13$^{Pru}$ without need of ligation to a known ligand for ubiquitination machinery. The generation of hRpn13$^{Pru}$ by proteasomal cleavage of full length hRpn13 (Fig. 7a, b) stymies our ability to definitively test

whether the loss of hRpn13$^{Pru}$ by **XL44** is proteasome dependent. However, PCLAF and RRM2 are depleted following **XL44** treatment by a proteasome dependent pathway (Figs. 7b, d, e). We also found this pathway to require the PCLAF KEN box (Fig. 7f). **XL44** also caused reduction of PTTG1, a downstream target of PCLAF[36] and PRR11[37]. PTTG1 has a KEN box and it is possible that **XL44** targets PTTG1 either indirectly through PCLAF or PRR11 or directly through a similar KEN box mechanism. Our TMT-MS data indicate reduction of KEN box proteins CDC25A and PRR11 following **XL44** treatment however lack of available antibodies prevented the validation of these results. hRpn13 targeting molecules or its depletion by RNAi disrupt cell cycle progression[40,41] and it is therefore possible that the loss of PCLAF and RRM2 (and perhaps also CDC25A and PRR11) is caused by a cell cycle defect. Examination of all APC/C substrates detected in **XL44**-treated RPMI 8226 cells by TMT-MS (Fig. 6a) however indicates only PCLAF, RRM2, and CDC25A were reduced by **XL44** (Supplementary Fig. 12), suggesting a more specific mechanism of action. Moreover, we find that RRM2 can be isolated from RPMI 8226 lysates in a pulldown assay with biotinylated **XL44** (Fig. 6h). Future experiments are needed to define whether this interaction involves other proteins and/or if the RRM2 KEN box region is used for binding to **XL44**.

Loss of hRpn13$^{Pru}$ by **XL44** is ubiquitin-independent, as its partial proteolysis to lower molecular weight species does not require UAE activity (Fig. 7b, c). These products however are stabilized by E1 inhibition, suggesting that complete depletion of hRpn13 by proteasomes requires ubiquitination (Fig. 7b, c). In the case of hRpn13, ubiquitin is not needed for its recruitment to the proteasome, as it binds to hRpn2[2,11,12] and the proteasome generates hRpn13$^{Pru}$ even in isolation of other cellular factors (Fig. 7a). Remaining questions however are how hRpn13$^{Pru}$ depletion is promoted by **XL44**. It is possible that **XL44** acts as a molecular glue to induce or enhance hRpn13$^{Pru}$ binding to an interaction partner. hRpn13 is present both on and off proteasomes[12,15,17] and an enhanced interaction through **XL44** could involve extra-proteasomal hRpn13 and/or compete with hRpn13 binding to the proteasome. Unlike carfilzomib (Fig. 7b) or **XL5-VHL-2**[21], **XL44** does not cause the accumulation of ubiquitinated hRpn13 (Fig. 7b). This finding supports a distinct clearance mechanism of **XL44** compared to **XL5-VHL-2**.

Knockdown of PCLAF in trRpn13 HCT116 cells impaired **XL44**-induced restriction of cell viability (Fig. 7g and Supplementary Fig. 11). PCLAF is required for cell stemness and plasticity of breast cancer cells and intestinal stem cells[42,43]. Lower PCLAF and RRM2 levels were associated with better survival for patients with myeloma (Fig. 6g). Moreover, RRM2 and CDC25A are among a 37 gene product signature for responsiveness to rapamycin and entinostat in myeloma and thereby prognostic of risk and survival[44]. **XL44**-induced apoptosis in myeloma cells therefore has the potential to increase the survival probability of myeloma patients. Notably, **XL44** is more drug-like, with

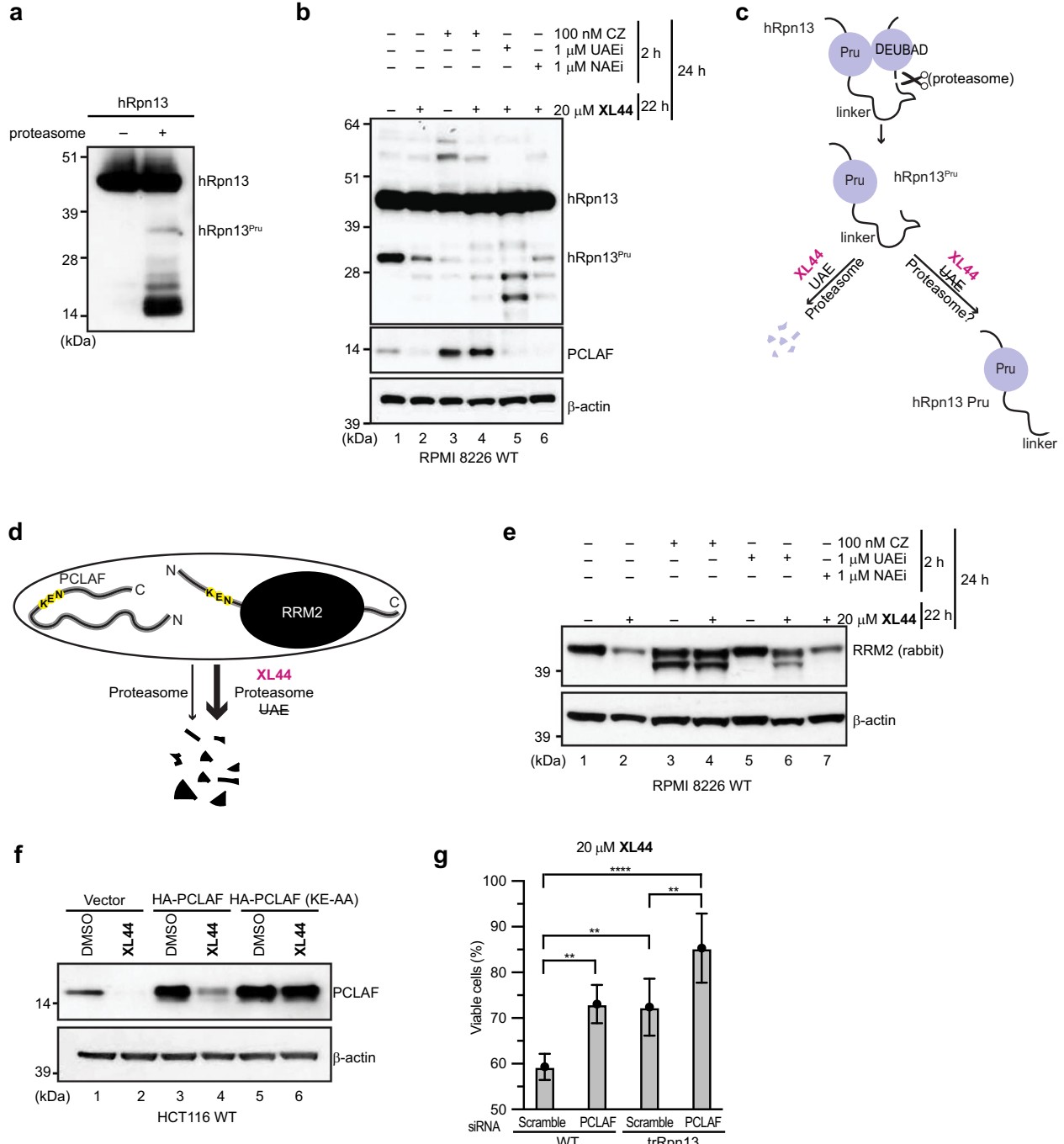

**Fig. 7 | XL44 dually targets hRpn13 and PCLAF. a** Immunoblots of hRpn13 from 640 nM hRpn13 incubated without or with 30 nM commercial 26 S proteasome at 37 °C for 10 min. **b, e** Immunoblots of hRpn13 **b**, PCLAF **b**, RRM2 **e**, and β-actin **b, e** from lysates of RPMI 8226 cells pre-treated for 2 h with 100 nM proteasome inhibitor carfilzomib (CZ), 1 μM UAE inhibitor MLN7243 (UAEi) or 1 μM NAE inhibitor MLN4924 (NAEi), followed by 22 h treatment with 20 μM **XL44** or DMSO. **c** Schematic representation of hRpn13 proteolysis (displayed as scissors) by proteasomes to generate hRpn13^Pru and its clearance by **XL44**. **XL44**-induced depletion of hRpn13^Pru requires UAE and proteasome activity, with hRpn13^Pru proteolysis followed by ubiquitin-dependent degradation. hRpn13 domains and intrinsically disordered regions are represented by purple circles and black lines, respectively. **d** Schematic representation of degradation of PCLAF and RRM2 by proteasomes, as accelerated by **XL44**. Intrinsically disordered region (bold line), domain (black

oval), and KEN box (yellow) are displayed, respectively. **f** Lysates immunoprobed for PCLAF and β-actin from HCT116 cells transfected with 1 μg empty vector (control), HA-PCLAF or HA-PCLAF (AA, with K78A and E79A substitutions), followed 48 h later by treatment with 20 μM **XL44** or DMSO for 24 h. **g** Cell metabolism measured by an MTT assay for HCT116 WT or trRpn13 cells transfected with 50 nM scramble or PCLAF siRNA for 48 h, followed by 24 h treatment with 20 μM **XL44** or DMSO (vehicle control). Data represent mean ± SD of n = 6 biological replicates. ****$P < 0.0001$; **$P < 0.004$; $P$ (WT Scramble vs. WT PCLAF) = 0.0018; $P$ (WT Scramble vs. trRpn13 Scramble) = 0.0029; $P$ (trRpn13 Scramble vs. trRpn13 PCLAF) = 0.0031; $P$ (WT Scramble vs. trRpn13 PCLAF) < 0.0001; 'one-way ANOVA' (Tukey's multiple comparisons test) in GraphPad Prism9. Viability is calculated as $(\lambda_{570})_{sample}/(\lambda_{570})_{control}*100$ (%). The experiments in 7a, 7b, 7e and 7 f were performed twice. Source data are provided as a Source Data file.

its small size compared to existing PROTACs[21]. It is also cell permeable, motivating further testing in preclinical trials for use in hRpn13[Pru]-producing cells, with upregulated PCLAF and RRM2, such as myeloma cells.

## Methods

### Covalent docking virtual screen

Covalent docking screens were conducted with the Schrödinger software by running on 8 CPUs of the National Institutes of Health Biowulf cluster supercomputer. A library of 18 compounds (Supplementary Table 1) was generated based on the chemical similarity of compounds to **XL5** by using Enamine's REAL database (https://enaminestore.com/search). The entire **XL5**-binding cleft of hRpn13 was used for the screen, including all hRpn13 residues in contact with **XL5**, as defined by the NMR structure of **XL5**-ligated hRpn13 Pru (PDB 7KXI). The reactive residue and reaction type for covalent docking was set up with hRpn13 residue Cys88 and Michael addition. Docking was performed and scored with the 'Thorough' option of 'Pose Prediction' in the Schrödinger Prime software package.

### Sample preparation

The DNA encoding hRpn13 Pru (1–150), hRpn13 or hRpn2 (940–953) was cloned to pRSET vector with His or glutathione S-transferase (GST) tags at the N-terminus followed by a PreScission protease cleavage site as described previously[12]. Plasmid expressing human PCLAF was generated commercially (GenScript) by inserting synthesized coding sequence with codon optimization for full length PCLAF between the BamHI and NotI restriction sites of pGEX-6p-1 with a GST tag at the N-terminus followed by a PreScission protease cleavage site. hRpn13 Pru, hRpn13 and hRpn2 (940–953) or PCLAF were expressed in *E. coli* BL21(DE3) pLysS or BL21 (DE3) cells (Invitrogen). Cells were grown at 37 °C to optical density at 600 nm of 0.6 and induced for protein expression by addition of isopropyl-β-D-thiogalactoside (0.4 mM) for 20 h at 17 °C or 4 h at 37 °C. The cells were harvested by centrifugation at 4550 g for 40 min, lysed by sonication, and cellular debris removed by centrifugation at 31,000 g for 30 min. The supernatant for hRpn13 Pru (1–150), hRpn13, hRpn2 (940–953) or PCLAF was incubated with Talon Metal Affinity resin (Clontech) for 1 h or Glutathione Sepharose 4B (GE Healthcare Life Sciences) for 3 h and the resin washed extensively with buffer A (20 mM sodium phosphate, 300 mM NaCl, 10 mM βME, pH 6.5). hRpn13 Pru, hRpn13 or PCLAF was eluted from the resin by overnight incubation with 50 units per mL of PreScission protease (GE Healthcare Life Sciences) in buffer B (20 mM sodium phosphate, 50 mM NaCl, 2 mM DTT, pH 6.5) whereas GST-hRpn2 (940-953) was eluted in buffer B containing 20 mM reduced L-glutathione. The eluent of GST-hRpn2 (940-953) was concentrated to 2 mL, buffer exchanged to buffer B, and then incubated with 50 units per mL of PreScission protease overnight to cleave the GST tag. Following affinity chromatography, hRpn13 Pru, hRpn2 (940-953), PCLAF or hRpn13 was subjected to size exclusion chromatography with a Superdex75 or Superdex200 column on an FPLC system for further purification. $^{15}$N ammonium chloride and $^{13}$C glucose were used for isotopic labeling.

### Acquisition of compounds

**XL44** (Enamine ID Z1497347974) was purchased from Enamine with purity >90%; Carfilzomib (Catalog No. S2853), **MLN7243** (Catalog No. S8341) and **MLN4924** (Catalog No. S7109) were purchased from Selleckchem with purity of 99.75%, 99.61% and 99.99%, respectively. **XL5B, XL44B, XL52-XL56, XL44**-$^{13}$C$_6$-CB were synthesized and described in Supplementary Note 1.

### NMR experiments

To screen for binding by $^{1}$H, $^{15}$N HSQC (pulse sequence: fhsqcf3gpph[45]) experiments, each small molecule was dissolved in DMSO-$d_6$ and

added to 20 μM $^{15}$N-labeled hRpn13 Pru (spectral width (ppm) of 13 ($^{1}$H) or 26 ($^{15}$N), transmitter frequency offset (ppm) of 4.7 ($^{1}$H) or 120 ($^{15}$N), acquisition time of 10.6 h with 256 scans per increment) or $^{15}$N-labeled PCLAF (spectral width (ppm) of 13 ($^{1}$H) or 28 ($^{15}$N), transmitter frequency offset (ppm) of 4.7 ($^{1}$H) or 118 ($^{15}$N), acquisition time of 10.4 h with 256 scans per increment) at a molar excess of equimolar, 2- or 10-fold as indicated in NMR buffer (20 mM sodium phosphate, 50 mM NaCl, 2 mM DTT, 10% DMSO-$d_6$, pH 6.5). NMR experiments were conducted at 10 °C or 25 °C and on Bruker Avance 600, 700, 800 or 850 MHz spectrometers equipped with cryogenically cooled probes. The $^{13}$C-half-filtered NOESY (pulse sequence: noesyhmqc3d19x1.x) spectrum was acquired with a 100 ms mixing time on a mixture of 0.4 mM $^{13}$C-labeled hRpn13 Pru and 0.4 mM unlabeled **XL44** (spectral width (ppm) of 13 ($^{1}$H) or 42 ($^{13}$C), transmitter frequency offset (ppm) of 4.8 ($^{1}$H) or 24 ($^{13}$C), acquisition time of 160.9 h with 32 scans per increment) or another mixture of 0.4 mM hRpn13 Pru and 0.4 mM **XL44** with the central benzene ring $^{13}$C-labeled (**XL44**-$^{13}$C$_6$-CB) (acquired without incrementing the $^{13}$C dimension, spectral width (ppm) of 13 ($^{1}$H) or 22 ($^{13}$C), transmitter frequency offset (ppm) of 4.8 ($^{1}$H) or 125 ($^{13}$C), acquisition time of 135.6 h with 512 scans) in NMR buffer. $^{19}$F (pulse sequence: zg, spectral width of 19.9 ppm, transmitter frequency offset of −120 ppm, acquisition time of 1.6 h with 5000 scans) spectra were acquired on samples of 50 μM **XL44** alone or with equimolar hRpn2 (940-953) or equimolar hRpn13 Pru as well as of mixtures of preincubated 50 μM **XL44** and equimolar hRpn13 Pru with 25 or 50 μM hRpn2 (940-953) added. All samples were dissolved in NMR buffer without or with DTT as indicated and at 10 °C. The $^{1}$H, $^{15}$N heteronuclear NOE experiment was acquired on 440 μM $^{15}$N-labeled hRpn13 Pru with a saturation time of 6 s. Data were processed by NMRPipe[46] and visualized with XEASY[47], MestReNova (https://mestrelab.com/) or TopSpin 3.6.5 (https://www.bruker.com/en/products-and-solutions/mr/nmr-software/topspin.html).

### Differential scanning fluorimetry

DSF experiments were performed on a Prometheus NT.48 instrument (NanoTemper Technologies, Germany) at 20 °C. 2 μM hRpn13 Pru was added to equal volume of serially diluted **XL44** or **XL5** in buffer C (20 mM sodium phosphate, 50 mM NaCl, 2 mM DTT, 10% DMSO, pH 6.5). Each sample was loaded into three capillaries of High Sensitivity grade (NanoTemper, cat # PR-C006) and the emission of intrinsic tryptophan fluorescence at 350 nm was monitored.

### LC-MS experiment

2 μM hRpn13 Pru or 4 μM PCLAF was incubated with 10-fold molar excess **XL44** or DMSO (vehicle control) in buffer D (20 mM sodium phosphate, 50 mM NaCl, pH 6.5) containing 0.2% DMSO for 2 h at 4 °C and the samples were then loaded onto a 6520 Accurate-Mass Q-TOF LC/MS system equipped with a dual electro-spray source, operated in the positive-ion mode. Acetonitrile was added to all samples to a final concentration of 10%. Mass Hunter Qualitative Analysis software (version B.07.00) with Bioconfirm Workflow was used for data analysis and deconvolution of mass spectra.

### Lentivirus transduction experiments

Synthesized *ADRM1* cDNA encoding hRpn13 (Twist Biosciences) was cloned by isothermal assembly[48] into pGMC00021 (Addgene) to generate pMG0784, which following a maxiprep (Thermo Scientific Fisher), was verified by nanopore sequencing (Poochon Scientific). Lentivirus was generated by co-transfecting 8 μg of pMG0784, 5.757 μg of psPAX2 (Addgene) and 3.133 μg of pMD.2 G (Addgene) by 30 μL of Lipofectamine 2000 into a 15 cm plate of 293 T cells, with supernatants collected at 48 and 72 h, filtered by Steriflip 0.45 μm (Millipore), and then concentrated by a LentiX concentrator (Takara) for 3 days as per manufacturer's protocol. Following centrifugation and re-suspension in ~400 μL of ice-cold PBS, 7.2 and 20 μL aliquots were made. 50,000 RPMI

8226 trRpn13-MM2 cells were transduced with two of the 20 μL aliquots and 48 h later, the cells were spun down at 300 g and re-suspended in 200 μL of fresh RPMI media with 10% FBS, 1% Penstrep, and 2 μg/mL puromycin. The suspension was kept in this media for 2-weeks for selection. One week later, the cells were collected and transferred to a single well of a 24 well plate, the next week collected and transferred to a single well of a 6 well plate, and the next week transferred to a T25 flask.

## Cell culture and antibodies

The RPMI 8226 (ATCC® CCL-155™), HS5-BMSCs (ATCC® CRL-3611™), HCT116 WT (ATCC® CCL-247™), and MCF7 (ATCC® HTB-22™) cell lines were purchased from the American Tissue Culture Collection; OVCAR-4 (SCC258) and OVCAR-5 (SCC259) were purchased from Millipore-Sigma; Human Fibroblast 1634 (HF-1634) cells were a generous gift from Dr. Douglas Lowy, RPMI 8226 trRpn13-MM2 or HCT116 trRpn13 cells were generated and described in our previous studies[21,40]. RPMI 8226, OVCAR-4, and OVCAR-5 cell lines were grown in RPMI-1640 media (ATCC® 30-2001™) with 0.25 U/mL insulin (OVCAR-4 only). HS5 and HF-1634 cells were cultured in DMEM media supplemented with 100 mg/mL penicillin and streptomycin 100 U/mL, (Gibco, Life Technologies). HCT116 cells were grown in McCoy's 5 A modified media (Thermo Fisher Scientific 16600082). MCF7 cells were cultured in Eagle's Minimum Essential Medium (ATCC, 30-2003) with 0.01 mg/ml human recombinant insulin. In all cases, the media was supplemented with 10% fetal bovine serum (Atlanta Biologicals) and growth occurred in a 37 °C humidified atmosphere of 5% $CO_2$. Antibodies (dilutions) used in this study include primary antibodies anti-hRpn13 (100-200) (Abcam ab157185, EPR11449(B), 1:5,000), anti-β-actin (Cell Signaling Technology 3700 s, 8H10D10, 1:6,000 or 1:10,000), anti-cleaved caspase-9 (Cell Signaling, 52873 s, E5Z7N, 1:500), anti-caspase-3 (Cell Signaling, 9662 s, 1:1000), anti-PCLAF (Santa Cruz, sc-390515 HRP, G-11, 1:1000 or 1:500), anti-PCLAF (Cell Signaling, 81533 s, D8E2Y, 1:1000), anti-PTTG1 (Cell Signaling, 13445 s, D2B6O, 1:1000), anti-RRM2 (Abcam ab57653, 1E1, 1:1000), anti-RRM2 (Abcam ab172476, EPR11820, 1:3000), anti-ubiquitin (P4D1) (Cell Signaling, 3936 s, P4D1, 1:1000), secondary antibodies anti-mouse (Sigma-Aldrich, A9917, 1:3000 or 1:4000), anti-rabbit (Life Technologies, A16110, 1:5000) antibodies.

## MTT assay

Cells were seeded at 8000 cells/well in 96-well plates and after 24 h, treated with 0.4% DMSO (as a vehicle control) or **XL44** or **XL52** at 2 μM, 4 μM, 6 μM, 8 μM, 12 μM, 16 μM or 20 μM concentration with DMSO maintained at 0.4%. Each condition was performed in triplicate or sextuplicate. After 48 h, 0.35 mg/mL MTT was added for 4 h of incubation. Stop solution (40% DMF, 10% SDS (W/V), 25 mM HCl, 2.5% acetic acid in $H_2O$) was added to the cells and incubated overnight. Absorbance at 570 nm was measured by using CLARIOstar (BMG LABTECH).

## MTS assay

Cell proliferation experiments were performed using Cell Titer 96 Aqueous One Solution Cell Proliferation Assay System (Promega, Madison, WI, USA). Briefly, cells were plated in quadruplicate on clear, flat-bottomed 96-well tissue culture plates (Corning Costar) and treated with 0.1 to 50 μM concentrations of **XL44**. After incubation for 48 h, MTS reagent was added directly to the wells and incubated for 1.5 h. The absorbance of MTS formazan was read at 490 nm (Omega 640 spectrophotometer, BMG Labtech, Cary, NC, USA). A blank measurement was taken for the absorbance of the wells with media only and subtracted accordingly.

## LDH assay

LDH activity in cell supernatants was assessed using CQUANT LDH Cytotoxicity Assay Kit (ThermoFisher). RPMI 8226 WT and RPMI 8226 trRpn13-MM2 cells were plated with increasing cell number to determine optimal cell density for the assay. Briefly, 25000 cells/well were plated in quadruplicate in 96-well plates and treated with 2.5 to 20 μM concentrations of **XL44** or solvent control (DMSO) for 24 h. Equal number of wells were plated for Spontaneous LDH activity and Maximum LDH activity controls. The chemical compound-mediated cytotoxicity assay was performed. % cytotoxicity was measured according to the manufacturer's protocol.

## siRNA knockdown assay

HCT116 WT or trRpn13 cells were seeded at 40,000 cells/mL with RPMI 1640 medium (no phenol red, Thermo Fisher Scientific 11835030) containing 10% fetal bovine serum and transfected with 50 nM scramble (Horizon Discovery, D-001810-10-05), PCLAF siRNA (Horizon Discovery, L-017672-01-0010) or RRM2 siRNA (Horizon Discovery, L-010379-00-0010) by using the Lipofectomine™ 300 transfection reagent for 48 h. After 48 h, the cells transfected with scramble (control) or PCLAF siRNA were seeded in a 10 cm dishes and treated with 0.8% DMSO or 20 μM **XL44** for 24 h or in parallel, seeded for the MTT assay into 96-well plates and treated with 0.4% DMSO or **XL44** at 10 μM or 20 μM for 24 h.

## Plasmids for transfection

Plasmids expressing HA-tagged PCLAF WT or PCLAF mutant AA (K78A, E79A) were generated commercially (GenScript) by inserting synthesized coding sequence for full length PCLAF WT (UniProt ID: Q15004) or mutant AA with residues lysine 78 and glutamic acid 79 mutated to alanine between the BamHI and NotI restriction sites of pcDNA3.1 + N-HA (GenScript, SC1317). Unmodified pcDNA3.1 + N-HA was used as empty vector (EV) control.

## Transfection

HCT116 WT cells ($8 \times 10^4$) were reverse transfected with 1 μg empty vector (EV), HA-PCLAF, HA-PCLAF (AA)-expressing plasmid by using lipofectamine 3000 (Thermo Fisher Scientific, L3000015) according to the manufacturer's instructions. After 48 h of transfection, cells were treated with 20 μM **XL44** or DMSO (a control) for 24 h before harvesting cells.

## XL44 treatment

Two or four million RPMI 8226 WT, trRpn13-MM2 or hRpn13-expressing trRpn13-MM2 cells were seeded separately in a T75 flask. After 48 h, the cells were treated with 0.8% DMSO (as a control, 0 h or 24 h) or 1, 2.5, 5, 7.5, 10, 15, 20 μM **XL44**, 1 μM **MLN7243** and 20 μM **XL44** for 24 h or 20 μM **XL44** for 1, 2, 3, 4, 5, 7, 8, 12 or 24 h, or cells were pretreated with 100 nM carfilzomib, 1 μM **MLN7243** or 1 μM **MLN4924** for 2 h and then 20 μM **XL44** added for another 22 h of co-treatment, as indicated.

## Biotin pulldown assay

RPMI 8226 WT cell lysates (2 mg) were preincubated with 50 μL pre-washed Dyna beads MyOne streptavidin T1 magnetic beads (Invitrogen 65602) for 1 h at 4 °C to remove non-specific biotinylated proteins in the cell lysate and then pre-cleared cell lysate was incubated with 40 μM biotin, **XL5B** or **XL44B** overnight at 4 °C. Next, the mixture was incubated for an additional 3 h at 4 °C with 50 μL pre-washed Dyna beads MyOne streptavidin T1 magnetic beads. Following three washes with 1% Triton-TBS lysis buffer, proteins bound to the Dyna beads MyOne streptavidin T1 magnetic beads were eluted by using 2x LDS with 100 mM DTT and analyzed by immunoblotting.

## Immunoblotting

Cells were collected and washed with PBS followed by flash freezing in liquid nitrogen and storage at −80 °C for future use. Cells were lysed in 1% Triton-TBS lysis buffer (50 mM Tris-HCl, pH 7.5, 150 mM NaCl, 1 mM

PMSF) supplemented with protease inhibitor cocktail (Roche). Total protein concentration was determined by bicinchoninic acid (BCA)) (Pierce). Protein lysates were prepared in 1x LDS (ThermoFisher, NP0007) buffer with 100 mM DTT and heating at 70 °C for 10 min, loaded onto 4–12% Bis-Tris polyacrylamide gels (Life Technologies), subjected to SDS–PAGE and transferred to Invitrolon polyvinylidene difluoride membranes (Life Technologies). The membranes were blocked in Tris-buffered saline with 0.1% Tween-20 (TBST) supplemented with 5% skim milk or 5% BSA, incubated with primary antibody, washed in TBST, incubated with secondary antibodies, and washed extensively in TBST. Pierce™ ECL Western Blotting Substrate (32106; Thermo Fisher Scientific) or Amersham™ ECL™ Prime Western Blotting Detection Reagent (cytiva) was used for antibody signal detection.

### hRpn13 degradation assay

640 nM hRpn13 was incubated without or with 30 nM commercial 26 S proteasome (Enzo, BML-PW9310) for 10 min at 37 °C in a buffer containing 50 mM Tris-HCl (pH 7.5), 5 mM MgCl$_2$ and 5 mM ATP. The reaction was stopped by adding LDS loading buffer with 100 mM DTT.

### TMT proteomic analysis

RPMI 8226 WT or trRpn13-MM2 cells were treated with 0.8% DMSO (as a control) or 20 μM **XL44** for 8 h. Cell pellets were lysed in EasyPrep Lysis buffer (Thermo Fisher Scientific) according to the manufacturer's protocol. Lysates were clarified by centrifugation and protein concentration was quantified by using a bicinchoninic acid protein estimation kit (Thermo Fisher Scientific). 20 μg of lysate was reduced, alkylated, and digested by addition of trypsin at a ratio of 1:50 (Promega) and incubating overnight at 37°C. For TMT labeling, 100 μg of TMTpro label (Thermo Fisher Scientific) in 100% acetonitrile was added to each sample. After incubating the mixture for 1 h at room temperature with occasional mixing, the reaction was terminated by adding 50 μL of 5% hydroxylamine, 20% formic acid. The peptide samples for each condition were pooled and clean-up was performed using the proprietary peptide clean up columns from the EasyPEP Mini MS Sample Prep kit (Thermo Fisher Scientific). The first dimensional separation of the peptides was performed by a Waters Acquity UPLC system coupled with a fluorescence detector (Waters, Milford, MA) and a 150 mm × 3.0 mm Xbridge Peptide BEM™ 2.5 μm C18 column (Waters, MA) operating at 0.35 mL/min. The dried peptides were reconstituted in 100 μL of mobile phase A solvent (3 mM ammonium bicarbonate, pH 8.0); mobile phase B was 100% acetonitrile (Thermo Fisher Scientific). The column was washed with mobile phase A for 10 min followed by gradient elution 0–50% B (10–60 min) and 50–75% B (60–70 min). Fractions were collected every minute and the resulting 60 fractions pooled into 24 fractions that were vacuum centrifuged to dryness and stored at −80 °C until analysis by mass spectrometry. The dried peptide fractions were reconstituted in 0.1% trifluoroacetic acid and subjected to nanoflow liquid chromatography (Thermo Ultimate™ 3000RSLC nano LC system, Thermo Fisher Scientific) coupled to an Orbitrap Eclipse mass spectrometer (Thermo Fisher Scientific). Peptides were separated by a low pH gradient with a 5–50% acetonitrile over 120 min in mobile phase containing 0.1% formic acid at 300 nL/min flow rate. For TMT analysis, the FAIMS-MS2-based approach was used. MS scans were performed in the Orbitrap analyzer at a resolution of 120,000 with an ion accumulation target set at 4e$^5$ and max IT set at 50 ms over a mass range of 350–1600 m/z. The FAIMS source was operated under standard resolution and four different compensation voltages (CVs) of -45, -60, -75, and -90 were used. Ions with determined charge states between 2 and 6 were selected for MS2 scans. A cycle time of 0.75 s was used for each CV and a quadrupole isolation window of 0.4 m/z was used for MS/MS analysis. An Orbitrap at 15,000 resolution with a normalized automatic gain control set at 250 followed by maximum injection time set as "Auto" with a normalized collision energy setting of 38 was used for MS/MS analysis. The node "Turbo

TMT" was switched on for high-resolution acquisition of TMT reporter ions. Acquired MS/MS spectra were searched against a human uniprot protein database using a SEQUEST HT and percolator validator algorithms in the Proteome Discoverer 2.4 software (Thermo Fisher Scientific). The precursor ion tolerance was set at 10 ppm and the fragment ions tolerance was set at 0.02 Da along with methionine oxidation included as a dynamic modification. Carbamidomethylation of cysteine residues and TMT16 plex (304.2071 Da) was set as a static modification of lysine and the N-termini of the peptide. Trypsin was specified as the proteolytic enzyme, with up to 2 missed cleavage sites allowed. Searches used a reverse sequence decoy strategy to control for the false peptide discovery and identifications were validated using percolator software. Only proteins with less than 50% co-isolation interference were used for quantitative analysis. Reporter ion intensities were adjusted to correct for the impurities according to the manufacturer's specification and the abundances of the proteins were quantified using the summation of the reporter ions for all identified peptides. The reporter abundances were normalized across all the channels to account for equal peptide loading. Data analysis and visualization were performed by Microsoft Excel or R.

### Crystallization of the hRpn13 Pru-XL44 complex

A mixture of 374 μM hRpn13 Pru and 763 μM ubiquitin dissolved in 20 mM HEPES (pH 7.5), 100 mM NaCl and 1 mM TCEP and 509 μM **XL44** dissolved in DMSO was crystallized by microbatch-under-oil at 4 °C with equivolume of 0.1 M citric acid pH 4.6 and 20% PEG4000. Before data collection crystals were cryo-protected in a reservoir solution containing 25% ethylene glycol.

### Data collection, processing, and structure determination

X-ray diffraction data was collected at the Southeast Regional Collective Access Team (SER-CAT) using beamline ID-22 (wavelength 1.0 Å) available at the Advanced Photon Source, Argonne National Laboratory equipped with an EIGER 16 M detector. The X-ray data sets were processed and scaled by SER-CAT auto-processing software and the structure solved by molecular replacement by using the program PHENIX[49] and the crystal structure of the hRpn13 Pru: ubiquitin: Rpn2 complex (PDB 5V1Y) as the search model[13]. All structural refinement was performed by using PHENIX and REFMAC5 (CCP4 package) with default parameters. The Matthews coefficient[50] indicated the presence of two molecules in the asymmetric unit. The model building was done manually in Coot[51]. The protein atoms were fit into the electron density contoured at 3.0σ and 1.0σ for the Fo-Fc and 2Fo-Fc maps, respectively, and then **XL44** was fit into the electron density near the Cys88 using the difference map contoured at 3.0σ. Water molecules were then fit into the electron density and refinement was performed. Electron density for the Leu33, Val38 and Lys42 side chain atoms was not defined. Further refinement of these residues as well as the methyl group of **XL44** was done by using the intermolecular NOE information obtained from the $^{13}$C half-filtered NOESY experiment as described above using the program XPLOR-NIH 3.4[52] followed by REFMAC from CCP4 package. All data collection and refinement statistics are detailed in the Table 1; the percentage of favored, allowed, and outlier residues in the Ramachandran plot is 98.34, 1.66 and 0.0, respectively.

### Modeling of XL44-ligated hRpn13 Pru with SS stereochemistry

An initial set of topology and parameter files for **XL44** were generated by ACPYPE and corrected to require the angles in the planar 6-membered rings to sum to 360°. **XL44** was covalently bonded to the hRpn13 Cys88 sulfur atom with S, S stereochemistry at **XL44** C12 and C13. With all other atoms frozen, NOE-derived distance restraints between hRpn13 Met31, Leu33, Val38, Lys42, Val85, Cys88, Val93, Val95 and **XL44** were used to calculate 100 model structures of **XL44**-ligated hRpn13 Pru with S, S chirality for **XL44** C12 and C13 by using XPLOR-

NIH 3.4. The lowest energy structure was selected for analysis and display.

## Mining MMRF data

MMRF CoMMpaSS data (release IA18) was accessed via the MMRF Researcher Gateway. After excluding patients with missing metadata, baseline bone marrow gene-level TPM estimates and overall survival measurements for $n = 754$ patients were used to fit Cox survival regression models for PCLAF and PTTG1. Model fitting was performed using the survival package in R[53]. Kaplan-Meier plots were constructed for the patients in the bottom and top quartiles of gene expression using the ggsurvplot function from the survminer package for R[52]. Using the same TPM-normalized gene expression data, co-expression was assessed between PCLAF and PTTG1 using Pearson correlation for all 754 patients' baseline bone marrow samples.

## Statistics and reproductivity

The values for $n$ represent replicates of biochemical assays displayed in Figs. 1c, 5c, d, e-h, 6d-f (bottom panel), 7g, Supplementary Figs. 6 and 11. For each figure, the number of replicates is indicated in the figure legend. $IC_{50}$ values in Fig. 5e, f were analyzed by using the equation '[Inhibitor] vs. normalized response - Variable slope' in GraphPad Prism9. $D_{ep}C_{50}$ and $d_{1/2}$ values in Figs. 5c, d and 6d-f were calculated by using the equation '[Inhibitor] vs. normalized response - Variable slope' ($Y = 100/(1 + (D_{ep}C_{50}/X)^{HillSlope})$; Y: normalized response; X: concentration of inhibitor; HillSlope: describes the steepness of the family of curves) (Figs. 5c and 6d) and 'One phase decay' ($Y = (Y0 - Plateau)*exp(-K*X) + Plateau$; Y: normalized response; Y0 is the Y value when X (time) is zero; Plateau is the Y value at infinite times, expressed in the same units as Y; K is the rate constant, expressed in reciprocal of the X axis time units; half-life is in the time units of the X axis computed as $\ln(2)/K$) (Figs. 5d and 6e, f) in GraphPad Prism9.

Mass Hunter Qualitative Analysis software (version B.07.00) with Bioconfirm Workflow was used to deconvolute mass spectra in Fig. 1d and Supplementary Fig. 9b. TMT analysis was performed in Microsoft Excel or R. Biophysical experiments and vitro assay including 2D NMR, DSF, MST, LC-MS and hRpn13 degradation assay were repeated at least once. Experiments using mammalian cells were repeated at least once. All replications were consistent. Only Experiments in 6a, 6d, 6 f were performed once.

## APC/C-dependent ubiquitination assays

The APC/C and its necessary ubiquitination machinery were purified largely, as previously described[54,55]. Purified PCLAF was fluorescently labeled via a sortase-dependent reaction containing 50 μM PCLAF, 1 mM *LPETGG peptide (* denotes a fluorescein group), 10 mM CaCl₂, and 1 μM sortase. The reaction was incubated on ice overnight and then fluorescent PCLAF was repurified via buffer exchange using a PD10 column (Cytiva) and size exclusion chromatography. To examine the effect of **XL44** on APC/C-dependent substrate ubiquitination, several different concentrations of components and conditions were tested and the results were consistent. In the panel shown, the reaction was carried out using pre-mixed 5 mM MgATP, 100 nM UBA1, 300 nM UBE2C, 0.25 mg/mL BSA, and 30 nM APC/C. Separately, 5 μM FZR1 and 5 μM fluorescent PCLAF were incubated for an hour on ice with 40 μM **XL44** or DMSO. The mixtures were then combined and equilibrated to room temperature. 100 μM ubiquitin was added to start the reaction which proceeded for 15 min before being quenched by SDS-containing buffer. After SDS-PAGE, the gel was subjected to fluorescent monitoring using a Typhoon Imager.

## Reporting summary

Further information on research design is available in the Nature Portfolio Reporting Summary linked to this article.

## Data availability

The structural coordinate for **XL44**-ligated hRpn13 Pru with ubiquitin in this study has been deposited in the Protein Data Bank (PDB) under accession code 8FTQ. The structural coordinates for **XL5**-ligated hRpn13 Pru mentioned in this study were previously deposited into the PDB under accession code 7KXI. The Enamine REAL database is publicly available under the EnamineStore (https://enaminestore.com/search). TMT-MS data have been deposited at MassIVE with accession number MSV000092929 (https://massive.ucsd.edu/ProteoSAFe/dataset.jsp?task=6aa6076dfa6f4322a9539b9eb71e81ff). Source data are provided in this paper.

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

## Acknowledgements

This work was supported by the Intramural Research Program of the CCR, NCI, NIH (1 ZIABC011490 to K.J.W. and FLEX award to K.J.W., R.E.S, B.A.M., and D.E.C.). D.L.B. and N.G.B. were supported by NIH T32GM008570 and R35GM128855, respectively. J.W.K. is supported by NIH, DHHS, under Contract No. 75N91019D00024. This work utilized the computational resources of the NIH HPC Biowulf cluster (http://hpc.nih.gov). H.M. and W.M. are supported in part by a grant from the NIH R01GM118474 and R01AI150478 and federal funds from the NCI, NIH, under contract 75N91019D00024. W.M. is supported in part by the NIH Office of Intramural Training and Education's Intramural AIDS Research Fellowship. C.D.S. is supported by the Intramural Research Program of the NIDDK, NIH. M.J.E. is supported by NIH R01GM120309.

## Author contributions

K.J.W. and X.L. conceived of the project. X.L. performed the virtual screen, NMR, DSF, hRpn13 degradation assays, all cell biology experiments except for the MTS and LDS assay experiments, which were done by S.G. and N.G.K. with supervision by B.A.M. and MTT assays in the Fig. 5f which were done by B.T.S. with supervision D.E.C.; The crystal structure of hRpn13 Pru-**XL44** was solved by M.C. and refined with NMR data by X.L. and M.C.; LC-MS experiments were acquired by M.D. and analyzed by X.L., M.C. and M.D.; Samples for TMT experiments were prepared by X.L.; S.C. performed TMT experiments and analysis with supervision by T.A.; V.S. synthesized **XL5B, XL44B, XL52-XL56, XL44**-$^{13}C_6$-CB, which were designed by R.S., K.J.W., and V.S.; C.D.S. generated modified topology and parameter files for **XL44**. MMRF data were mined by B.A.M. and V.K.H.; W.M performed the heteronuclear NOE experiments with supervision by H.M.; D.L.B. performed APC/C-dependent ubiquitination assays with supervision by N.G.B. X.L. performed $^{19}F$ NMR experiments with assistance from J.W.K.; M.E.C. and R.C generated the hRpn13-expressing trRpn13-MM2 cell line. M.J.E. contributed to interpretation of TMT-data; K.J.W., X.L., and M.C. wrote the manuscript with contributions from all authors.

## Funding

## Competing interests

K.J.W., X.L., M.C., V.R.S., and R.E.S. declare filing of a patent application (U.S. Provisional Patent Application No. 63/539,663) covering **XL44** and **XL44** derivatives described in this manuscript. The other authors declare no competing interests.

## Additional information

[1]Protein Processing Section, Center for Structural Biology, Center for Cancer Research, National Cancer Institute, National Institutes of Health, Frederick, MD, USA. [2]Chemistry and Synthesis Center, National Heart, Lung, and Blood Institute, National Institutes of Health, Bethesda, MD, USA. [3]Laboratory of Cancer Biology and Genetics, National Cancer Institute, Bethesda, MD, USA. [4]Radiation Oncology Branch, Center for Cancer Research, National Cancer Institute, National Institutes of Health, Bethesda, MD, USA. [5]Protein Characterization Laboratory, Cancer Research Technology Program, Frederick National Laboratory for Cancer Research, Leidos Biomedical Research, Inc., Frederick, MD, USA. [6]Cancer Innovation Laboratory, Frederick National Laboratory for Cancer Research, Frederick, MD, USA. [7]Genome Modification Core, Frederick National Laboratory for Cancer Research, Frederick, MD, USA. [8]Computational Biomolecular Magnetic Resonance Core, Laboratory of Chemical Physics, National Institute of Diabetes and Digestive and Kidney Diseases, National Institutes of Health, Bethesda, MD, USA. [9]Biophysics Resource, Center for Structural Biology, Center for Cancer Research, National Cancer Institute, National Institutes of Health, Frederick, MD, USA. [10]Department of Biochemistry and Biophysics and Lineberger Comprehensive Cancer Center, The University of North Carolina at Chapel Hill, Chapel Hill, NC, USA. [11]Basic Science Program, Leidos Biomedical Research Inc., NMR Facility for Biological Research, Center for Structural Biology, National Cancer Institute, National Institutes of Health, Frederick, MD, USA. [12]Department of Pharmacology and Lineberger Comprehensive Cancer Center, The University of North Carolina at Chapel Hill, Chapel Hill, NC, USA. ✉e-mail: kylie.walters@nih.gov

