## [Peer Review File · Nature Communications]

Reviewers' Comments:

Reviewer #1:

Remarks to the Author:

The manuscript by Lu et al reports several surprising and intriguing findings with important implications for the treatment of cancer. The authors build on their previous discovery of an hRpn13 covalent ligand XL5 that was attached to a VHL recruiter to make an hRpn13 degrader. Here the authors report a second generation ligand XL44 that induces degradation on its own, without attachment of a VHL recruiter. The binding of XL44 to Rpn13 is thoroughly characterized including nmr and x-ray crystallography experiments. The authors also show that XL44 induces apoptosis. Lastly they identify several other proteins down regulated by XL44. The work is rigorous and sound. The results are very interesting and provoke additional questions that will undoubtedly be addressed in the future. A couple suggestions to improve the manuscript:

Major points

1. Does XL44 interact directly with PCLAF, PTTG1 and RRM2? Have the authors probed XL44B associated proteins (Fig. 1f) for the presence of these proteins?
2. Perhaps this reviewer missed it, but the syntheses and compound characterization for XL44B etc. appear to be missing.
3. Line 496-498, hRpn13 is recruited to the proteasome by hRpn2, but XL44 competes with hRpn2, and should therefore prevent recruitment, so how does XL44 increase hRpn13 degradation?

Minor points

4. A cartoon summarizing the interactions of hRpn13, hUCHL5 and hRpn2 would be very helpful to a general reader.
5. Fig 5: please give actual equations used to fit Fig. 5b and 5c in the Methods. Also please indicated that the XL5 values are provided for comparison and include citation in the legend
6. p15, line 330: XL44 clearly decreases the viability of multiple cancer cell lines, but have only demonstrated that this effect is hRpn13-independent in one cell line, so this conclusion should be toned down.
7. Line 490, typo "CDC25A is reduced by..."

Reviewer #2:

Remarks to the Author:

The ubiquitin-binding proteasomal subunit Rpn13 is a potential anti-cancer therapeutic target. In this manuscript, the Walters lab uses virtual screening to identify a small molecule referred to as XL44, which can bind the Pru domain of Rpn13. XL44 exhibits greater potency than the previously reported Rpn13 binder XL5 and a series of elegant structural studies shows that XL44 binds the same groove of Rpn13-Pru normally occupied by the proteasomal subunit Rpn2. Once the studies pivot to elucidating the cellular mode of action, however, it becomes unclear whether Rpn13 has anything to do with the function of XL44. Using quantitative proteomics, the authors find several proteins unrelated to Rpn13 (e.g., PCLAF, RRM2, CDC25A) whose levels are reduced in the presence of XL44. The reduction in abundance appears to be due to induced degradation through the proteasome in a ubiquitin-independent manner. Perhaps the most compelling data in terms of understanding the function of XL44 is the observation that knocking down PCLAF levels using siRNA phenocopies the effects of XL44 on cell viability. That said, it is unclear whether PCLAF or the other proteins, e.g., RRM2 and CDC25A, are direct targets of XL44. To summarize, the in vitro characterization of XL44 binding to the PRU domain of Rpn13 is clear and convincing, but the data suggesting the cellular mode of action is related to Rpn13 is much less compelling.

Specific concerns

Figure 1f – Is Rpn13 the primary target of XL44B? Was PCLAF, RRM2, or CDC25A detected in the resin-bound fraction? It is plausible that the induced degradation precludes detection of XL44 targets. Quantitative proteomics comparing the levels of XL44B-enriched proteins to biotin-enriched proteins from cells treated with a proteasome inhibitor could help identify the principal

targets of XL44 in cells.

Figure 5a-b – Does expression of Rpn13 or Rpn13-PRU in trRpn13-MM2 cells rescue the WT phenotype with respect to caspase 9 cleavage? In the absence of these data, it is hard to claim XL44 induces apoptosis in a Rpn13-dependent manner, as the trRpn13-MM2 proteome could be very different at baseline relative to WT cells. Moreover, how is apoptosis induced by the loss of Rpn13-PRU when the latter is a minor species compared to full-length Rpn13, which is not degraded by XL44?

Figure 6 (related to the first point about Figure 1f) – Does XL44 interact with PCLAF or RRM2? If so, how does the strength of the interaction compare to Rpn13?

Figure 7 – In the absence of the rescue experiments mentioned above in relation to Figure 5, the claim that XL44 restricts cell viability by dually targeting Rpn13-PRU and PCLAF is not substantiated. Based on the data, it is plausible the cellular effects of XL44 could be entirely due to PCLAF.

Reviewer #3:

Remarks to the Author:

The article "A structure-based designed small molecule depletes hRpn13Pru and a select group of KEN box proteins" by Xiuxiu Lu et.al used structure guided virtual screening, to identify an hRpn13 binder (XL44) and solved its structure ligated to hRpn13 Pru. They show depletion of hRpn13Pru up on XL44 treatment in myeloma cells. Global proteomics identified PCNA clamp associated factor PCLAF and ribonucleoside-diphosphate reductase subunit M2 (RRM2) depletion by XL44 treatment. These findings open up new promise for XL44 as a potential therapeutic agent. This study is undoubtedly a valuable contribution to the scientific community. However, there are some major concerns that need to be addressed before this research can be fully accepted.

Discuss the rationale for choosing the 18 compounds in your virtual library, and the parameters used for ranking these compounds. In the text line 137-140 mentioned that XL44 is an isomeric mixture, did the authors explore these two stereoisomers binding profile with the hRpn13 and its consequences.

#The streptavidin bound biotinylated XL44 immunoprecipitation isolates hRpn13 is showed by WB. The mass spectrometry analysis will give clear binding profile of the XL44, and it is important to show that XL44 specifically targets hRpn13, and not the other proteins in the cell lysate. If other proteins are also observed binding to XL44, the authors should discuss it.

#The authors should provide justification for the use of ubiquitin for the crystal structure and whether addition of the ubiquitin is necessary and how it affect the interpretation. The author should also justify whether resolution at 2.01 Å. is sufficient for accuracy and completeness of the structure and how does it compare to typical resolution values in structural biology?

#The authors can provide insights into the mechanism by which XL44 induces apoptosis in RPMI 8226 cells. How does this depletion in hRpn13Pru relate to the induction of apoptosis. Similarly, the author has mentioned that XL44 reduce cell viability in an hRpn13 independent manner specifically in cancer cell lines compared to non-cancerous. More insight in mechanism is needed.

#The details of the tandem mass tag mass spectrometry (TMT-MS) experiment are missing in the result section. How many biological replicates were used. The number of the identified proteins. Reduction of five proteins were observed is mentioned in the results. What was the criteria. Are there is only five proteins were altered. Whether hRpn13 also detected in the TMT-MS. The proteomics data needs to deposit in the public repositories.

#The XL44 induced depletion of the PCLAF, PRR11 and PTTG1 were independent of the hRpn13 degradation. The author can give more insight by elaborating XL44 mechanism, if it works as

molecular glues then which E3 ligases are involved.

REVIEWER COMMENTS

Reviewer #1 (Remarks to the Author):

The manuscript by Lu et al reports several surprising and intriguing findings with important implications for the treatment of cancer. The authors build on their previous discovery of an hRpn13 covalent ligand XL5 that was attached to a VHL recruiter to make an hRpn13 degrader. Here the authors report a second generation ligand XL44 that induces degradation on its own, without attachment of a VHL recruiter. The binding of XL44 to Rpn13 is thoroughly characterized including nmr and x-ray crystallography experiments. The authors also show that XL44 induces apoptosis. Lastly they identify several other proteins down regulated by XL44. The work is rigorous and sound. The results are very interesting and provoke additional questions that will undoubtedly be addressed in the future. A couple suggestions to improve the manuscript:

We thank this Reviewer for their enthusiasm and critical reading of our manuscript. We have integrated all suggestions into the manuscript and agree that it's improved as a consequence.

Major points

1. Does XL44 interact directly with PCLAF, PTTG1 and RRM2? Have the authors probed XL44B associated proteins (Fig. 1f) for the presence of these proteins?

We thank the Reviewer for raising these important questions. In response, we have now tested and found that XL44 directly interacts with RRM2, but not PCLAF. These new data (shown below) are included as Fig. 6h and Supplementary Fig. 9 and involve immunoblots of XL44B-associated proteins (Fig. 6h), NMR (Supplementary Fig. 9a) and mass spectrometry (Supplementary Fig. 9b).

Fig. 6h, Immunoblots with antibodies against RRM2 (1 or 30 min) or PCLAF of streptavidin-bound **XL44B** or biotin (negative control) following incubation with RPMI 8226 cell lysates and subsequent wash steps to remove unbound proteins (top panels). Corresponding input from the lysates are included (lower panels).

Supplementary Fig. 9 XL44 does not bind to PCLAF. **a**, ^1H , ^{15}N HSQC spectra of 20 μM PCLAF (black) and with 10-fold molar excess of **XL44** (orange) in NMR buffer. The spectra were recorded at 700 MHz and 25°C. **b**, 4 μM purified PCLAF (MW: 12397.95 g/mol) was incubated with DMSO (a vehicle control, left panel) or 40 μM **XL44** (right panel) for 2 hours at 4°C and the samples subjected to LC-MS analysis to detect the formation of **XL44** adducts. UV spectra are displayed for each sample in **b**. Samples eluted between 1.74 and 2.01 minutes (dashed rectangle) were analyzed to generate MS spectra as indicated.

2. Perhaps this reviewer missed it, but the syntheses and compound characterization for XL44B etc. appear to be missing.

This information was included in Supplementary Note 1.

3. Line 496-498, *hRpn13* is recruited to the proteasome by *hRpn2*, but **XL44** competes with *hRpn2*, and should therefore prevent recruitment, so how does **XL44** increase *hRpn13* degradation?

In the original submission (line 144-158) we wrote: “1D ^{19}F NMR experiments indicate that **XL44** does not bind to *hRpn2*-bound *hRpn13* Pru, thus **XL44** cannot compete with *hRpn2*.” *hRpn13* is known to also exist outside of proteasomes and it may be this population of *hRpn13* that is targeted by **XL44** for degradation. We have now added this point to the revised manuscript along with citations of literature previously reporting *hRpn13* to be present also off proteasomes. Specifically, we added the text in red: “This experiment indicates that **XL44** does not bind to *hRpn2*-bound *hRpn13* Pru and that **XL44** cannot compete with *hRpn2*. *hRpn13* is present both on and off proteasomes^{12,15,17} and this data suggests that **XL44** targets extra-proteasomal *hRpn13*.”

Minor points

4. A cartoon summarizing the interactions of *hRpn13*, *hUCL5* and *hRpn2* would be very helpful to a general reader.

In the revised manuscript, we have now included a cartoon at the bottom of Fig. 5a, as suggested.

5. Fig 5: please give actual equations used to fit Fig. 5b and 5c in the Methods. Also please indicated that the XL5 values are provided for comparison and include citation in the legend.

In the revised manuscript, we now include the actual equations used to fit Fig. 5c (original Fig. 5b) and 5d (original Fig. 5c) in the Methods. We also now include the citation for the XL5-VHL-2 values in the legend.

6. p15, line 330: XL44 clearly decreases the viability of multiple cancer cell lines, but have only demonstrated that this effect is hRpn13-independent in one cell line, so this conclusion should be toned down.

This is a very good point and we agree that perhaps this effect may not be true in other cell lines. We have therefore altered the manuscript to change the subheading to “**XL44 restricts the viability of multiple cancer cell lines**” and also specified “**in trRpn13-MM2 cells**” in the text.

7. Line 490, typo “CDC25A is reduced by...”

We thank Reviewer for pointing this out; we have corrected this typo in the revised manuscript.

Reviewer #2 (Remarks to the Author):

The ubiquitin-binding proteasomal subunit Rpn13 is a potential anti-cancer therapeutic target. In this manuscript, the Walters lab uses virtual screening to identify a small molecule referred to as XL44, which can bind the Pru domain of Rpn13. XL44 exhibits greater potency than the previously reported Rpn13 binder XL5 and a series of elegant structural studies shows that XL44 binds the same groove of Rpn13-Pru normally occupied by the proteasomal subunit Rpn2. Once the studies pivot to elucidating the cellular mode of action, however, it becomes unclear whether Rpn13 has anything to do with the function of XL44. Using quantitative proteomics, the authors find several proteins unrelated to Rpn13 (e.g., PCLAF, RRM2, CDC25A) whose levels are reduced in the presence of XL44. The reduction in abundance appears to be due to induced degradation through the proteasome in a ubiquitin-independent manner. Perhaps the most compelling data in terms of understanding the function of XL44 is the observation that knocking down PCLAF levels using siRNA phenocopies the effects of XL44 on cell viability. That said, it is unclear whether PCLAF or the other proteins, e.g., RRM2 and CDC25A, are direct targets of XL44. To summarize, the in vitro characterization of XL44

binding to the PRU domain of Rpn13 is clear and convincing, but the data suggesting the cellular mode of action is related to Rpn13 is much less compelling.

We thank the Reviewer for their careful reading and thoughtful response to our manuscript. In response to these concerns, we have performed an add-back experiment in which hRpn13 is reintroduced to the *trRpn13-MM2* cell line. We hope that the Reviewer agrees that this experiment solidifies the targeting of hRpn13 for apoptosis. We also further characterize the interaction between PCLAF and RRM2 and explicitly note that out of 6469 proteins, only five additional ones (including PCLAF and RRM2) were identified as reduced by XL44. We hope the Reviewer agrees that this level of specificity is reasonable for a compound that has not yet been through thorough medicinal chemistry optimization.

Specific concerns

Figure 1f – Is Rpn13 the primary target of XL44B? Was PCLAF, RRM2, or CDC25A detected in the resin-bound fraction? It is plausible that the induced degradation precludes detection of XL44 targets. Quantitative proteomics comparing the levels of XL44B-enriched proteins to biotin-enriched proteins from cells treated with a proteasome inhibitor could help identify the principal targets of XL44 in cells.

From 6469 detected proteins, TMT-MS analyses indicate only PCLAF, PRR11, RRM2, RBIS, and CDC25A to be reduced by XL44. As we note in our response to Reviewer 1 (point 1), we have now tested and found that XL44 interacts directly with RRM2, but not PCLAF. These new data are shown above and included in the revised manuscript as Fig. 6h and Supplementary Fig. 9.

We agree that quantitative proteomics comparing the levels of XL44B-enriched proteins to biotin control would be a nice addition – and expected for a kinase for example. However, as our structure demonstrates, XL44-ligated hRpn13 binds to ubiquitin (Fig. 2a) and therefore treatment with a proteasome inhibitor (as suggested) would likely cause many proteins to be pulled out by ubiquitin interaction with XL44B-ligated hRpn13. To test whether our concern is legitimate, we immunoprobed for ubiquitin among XL44B-enriched proteins. Unfortunately as expected, ubiquitinated proteins were detected. This data and text stating the consequential limitation of using this approach for identification of additional XL44 targets is included as **revised Fig. 1f** and shown below.

Fig. 1f, Immunoblots with antibodies against hRpn13 (left panel) or ubiquitin (right panel) of RPMI 8226 lysates (bottom) or following a pulldown experiment with streptavidin-bound biotin or **XL44B** (top). For the pulldown, lysates from RPMI 8226 cells were incubated with streptavidin bound by biotin or XL44B and washed to remove unbound proteins (top).

Figure 5a-b – Does expression of Rpn13 or Rpn13-PRU in trRpn13-MM2 cells rescue the WT phenotype with respect to caspase 9 cleavage? In the absence of these data, it is hard to claim XL44 induces apoptosis in a Rpn13-dependent manner, as the trRpn13-MM2 proteome could be very different at baseline relative to WT cells. Moreover, how is apoptosis induced by the loss of Rpn13-Pru when the latter is a minor species compared to full-length Rpn13, which is not degraded by XL44?

To address this concern, we reintroduced hRpn13 into RPMI 8226 trRpn13 MM2 cells by using a lentiviral approach. Immunoblotting of the lysates revealed the presence of both reintroduced hRpn13 full length protein as well as its proteolytic hRpn13^{Pru} product. Importantly, XL44 treatment in this add-back cell line caused reduction of hRpn13^{Pru} and induced cleaved caspase-3. This new experiment indicates that hRpn13 is required and sufficient for XL44-induced apoptosis. We have now included this new data in the revised manuscript as Fig. 5b.

Fig 5b: Lysates from RPMI 8226 WT (WT), trRpn13-MM2 or trRpn13-MM2 with reintroduced hRpn13 (trRpn13-MM2/hRpn13) following 24 hour treatment with 20 μ M **XL44** or DMSO (vehicle control) immunoprobed for cleaved caspase-3, hRpn13 (short and long exposure), or β -actin (as a loading control).

Figure 6 (related to the first point about Figure 1f) – Does XL44 interact with PCLAF or RRM2? If so, how does the strength of the interaction compare to Rpn13?

As noted above in our response to Reviewer 1, we have now tested and found that XL44 directly interacts with RRM2, but not PCLAF. This new data is included as **revised Fig. 6h** and **Supplementary Fig. 9**.

Figure 7 – In the absence of the rescue experiments mentioned above in relation to Figure 5, the claim that XL44 restricts cell viability by dually targeting Rpn13-PRU and PCLAF is not substantiated. Based on the data, it is plausible the cellular effects of XL44 could be entirely due to PCLAF.

Happily, we were able to do the add-back experiment (**revised Fig. 5b**), which does reveal that hRpn13 is sufficient to rescue XL44-induced apoptosis.

Reviewer #3 (Remarks to the Author):

The article “A structure-based designed small molecule depletes hRpn13Pru and a select group of KEN box proteins” by Xiuxiu Lu et.al used structure guided virtual screening, to identify an hRpn13 binder (XL44) and solved its structure ligated to hRpn13 Pru. They show depletion of hRpn13Pru upon XL44 treatment in myeloma cells. Global proteomics identified PCNA clamp associated factor PCLAF and ribonucleoside-diphosphate reductase subunit M2 (RRM2) depletion by XL44 treatment. These findings open up new promise for XL44 as a potential therapeutic agent. This study is undoubtedly a valuable contribution to the scientific community. However, there are some major concerns that need to be addressed before this research can be fully accepted.

We thank the Reviewer for their enthusiasm and hope that they agree that we have addressed the major concerns.

Discuss the rationale for choosing the 18 compounds in your virtual library, and the parameters used for ranking these compounds.

In the revised manuscript, we updated the “Covalent docking virtual screen” section of Methods to include the URL of the Enamine database from where the 18 compounds of our virtual library were derived. In addition, we more clearly note that the docking scores were generated by the Schrödinger Prime software package.

In the text line 137-140 mentioned that XL44 is an isomeric mixture, did the authors explore these two stereoisomers binding profile with the hRpn13 and its consequences.

Yes, this information was included in the original manuscript and data provided in Supplementary Fig. 1a and 1b. The two stereoisomers behave equivalently.

The streptavidin bound biotinylated XL44 immunoprecipitation isolates hRpn13 is showed by WB. The mass spectrometry analysis will give clear binding profile of the XL44, and it is important to show that XL44 specifically targets hRpn13, and not the other proteins in the cell lysate. If other proteins are also observed binding to XL44, the authors should discuss it.

Reviewer 2 (point 1) made a similar suggestion – namely to use the biotinylated XL44 to pull out all possible interactors; however, as we show above, hRpn13 binds to ubiquitinated proteins which are therefore also co-enriched in a pulldown experiment. We now demonstrate this effect and include the associated data as Fig. 1f in the revised manuscript. Consistent with these findings, our structure indicates that XL44-bound hRpn13 binds to ubiquitin.

The authors should provide justification for the use of ubiquitin for the crystal structure and whether addition of the ubiquitin is necessary and how it affect the interpretation.

The author should also justify whether resolution at 2.1 Å. is sufficient for accuracy and completeness of the structure and how does it compare to typical resolution values in structural biology?

We attempted to crystalize hRpn13-bound XL44 without ubiquitin but were unsuccessful. We have now made this technical limitation clearer in the manuscript. Moreover, we also highlight that our structure indicates XL44-hRpn13 as competent for binding to ubiquitinated proteins. Resolution of 2.1 Å is considered of high quality – indeed all the high impact cryo-EM structures are of lower resolution. However, as we note in the manuscript, we also used NMR data to define XL44 binding to hRpn13 sidechain atoms beyond that possible by the x-ray data – our merger of NMR data with x-ray crystallography is indeed a strength of this manuscript and may set a new standard as many deposited ligand bound structures suffer from poor density at the contact site. This point is made in Figure 2, 3 and Supplementary Fig. 3. We also changed the title of the Figure 3 caption to make this point clearer.

The authors can provide insights into the mechanism by which XL44 induces apoptosis in RPMI 8226 cells. How does this depletion in hRpn13^{Pr} relate to the induction of apoptosis. Similarly, the author has mentioned that XL44 reduce cell viability in an hRpn13 independent manner specifically in cancer cell lines compared to non-cancerous. More insight in mechanism is needed.

In response to this and concerns raised by Reviewer 2, we have now included an add-back experiment in the revised manuscript as Fig. 5b. We reintroduced hRpn13 into RPMI 8226 trRpn13-MM2 cells by using a lentiviral approach. Immunoblotting lysates following this reintroduction indicated the presence of hRpn13 full length protein and its proteolysis to hRpn13^{Pr}. XL44 treatment caused reduction of reintroduced hRpn13^{Pr} and elevated cleaved caspase-3, indicating induction of apoptosis. This new experiment thus indicates that hRpn13 is required and sufficient for XL44-induced apoptosis.

The details of the tandem mass tag mass spectrometry (TMT-MS) experiment are missing in the result section. How many biological replicates were used. The number of the identified proteins. Reduction of five proteins were observed is mentioned in the results. What was the criteria. Are there is only five proteins were altered. Whether hRpn13 also detected in the TMT-MS. The proteomics data needs to deposit in the public repositories.

We have now included more details for the tandem mass tag mass spectrometry (TMT-MS) experiments and deposited the data at MassIVE with accession number MSV000092929. The red text was added to the manuscript address these concerns.

“We next performed tandem mass tag mass spectrometry (TMT-MS) on lysates from RPMI 8226 WT cells treated for 8 hours with 20 μM **XL44** or DMSO (vehicle control) **in triplicate. 6469 proteins were detected by this analysis, with five proteins identified to be ≥ 0.5-fold reduced and with p-values < 0.001**; these include PCLAF, RRM2, proline rich

11 (PRR11), ribosomal biogenesis factor (RBIS), and cell division cycle 25A (CDC25A) (Fig. 6a).”

The XL44 induced depletion of the PCLAF, PRR11 and PTTG1 were independent of the hRpn13 degradation. The author can give more insight by elaborating XL44 mechanism, if it works as molecular glues then which E3 ligases are involved.

We believe that XL44 likely acts as a molecular glue for these KEN box proteins, possibly through FZR1. However, testing this hypothesis is beyond the scope of this already extensive study. We plan to pursue this mechanistic question further but to do so well will require a considerable amount of time.

Reviewers' Comments:

Reviewer #1:

Remarks to the Author:

The authors have addressed all of my previous concerns. Recommend publication.

Reviewer #2:

Remarks to the Author:

The authors have adequately addressed my concerns.

Reviewer #3:

Remarks to the Author:

The article titled "A Structure-Based Designed Small Molecule Depletes hRpn13Pru and a Select Group of KEN Box Proteins," authored by Xiuxiu Lu et al., has addressed the majority of the reviewers' concerns by presenting additional experimental data. These new findings significantly bolster the claims made by the author in the manuscript. However, there are still a few remaining concerns that need to be addressed before the article can be accepted.

1. Upon the recommendations of the two reviewers, it has been proposed to conduct XL44 pulldown followed by mass spectrometry experiments. It is evident that the application of a proteasomal inhibitor significantly enhances the ubiquitination profile, aligning with the findings presented by the authors. This heightened ubiquitination is not anticipated to interfere with the mass spectrometry analysis. Consequently, it remains prudent to proceed with immunoprecipitation-mass spectrometry (IP-MS) to obtain a comprehensive binding profile of the compound. Another alternative suggestion, especially if the author remains concerned about the impact of ubiquitination due to the proteasomal inhibitor, is to consider conducting the pulldown at an earlier time point of XL44 without proteasomal treatment.

2. It is essential to conduct a compound washout treatment to determine the timeframe within which the depletion of hRpn13Pru is restored after discontinuation of XL44. Obtaining this data is crucial for gaining deeper insights into the dynamics of the depletion process.

Reviewer #1 (Remarks to the Author):

The authors have addressed all of my previous concerns. Recommend publication.

We thank the Reviewer for their enthusiasm towards the revised version of the manuscript and their much-valued recommendations.

Reviewer #2 (Remarks to the Author):

The authors have adequately addressed my concerns.

We are grateful to this Reviewer for their enthusiasm towards our revised manuscript and for helping us to improve it.

Reviewer #3 (Remarks to the Author):

The article titled "A Structure-Based Designed Small Molecule Depletes hRpn13Pru and a Select Group of KEN Box Proteins," authored by Xiuxiu Lu et al., has addressed the majority of the reviewers' concerns by presenting additional experimental data. These new findings significantly bolster the claims made by the author in the manuscript.

We thank the Reviewer for their enthusiasm towards our revised manuscript.

However, there are still a few remaining concerns that need to be addressed before the article can be accepted.

1. Upon the recommendations of the two reviewers, it has been proposed to conduct XL44 pulldown followed by mass spectrometry experiments. It is evident that the application of a proteasomal inhibitor significantly enhances the ubiquitination profile, aligning with the findings presented by the authors. This heightened ubiquitination is not anticipated to interfere with the mass spectrometry analysis. Consequently, it remains prudent to proceed with immunoprecipitation-mass spectrometry (IP-MS) to obtain a comprehensive binding profile of the compound. Another alternative suggestion, especially if the author remains concerned about the impact of ubiquitination due to the proteasomal inhibitor, is to consider conducting the pulldown at an earlier time point of XL44 without proteasomal treatment.

The experiment in Figure 1f where biotinylated XL44 pulls out hRpn13 with bound ubiquitinated proteins was performed **without** proteasome inhibitors. IP-MS would identify any ubiquitinated protein bound to hRpn13 and require a great deal of additional experiments to test whether any of these proteins are valid XL44 interactors. We believe that the rescue experiment provides firm evidence that hRpn13 is essential for XL44-induced apoptosis.

2. It is essential to conduct a compound washout treatment to determine the timeframe within which the depletion of hRpn13Pru is restored after discontinuation of XL44. Obtaining this data is crucial for gaining deeper insights into the dynamics of the depletion process.

The suggested experiment may inform on the dynamics of hRpn13^{Pru} recovery (by proteasome cleavage of the full length hRpn13 protein) but would not provide insight into the dynamics of the depletion process. Fig. 5d provides insight into the dynamics of the depletion, by showing a time course following XL44 treatment and half-life value for XL44-induced depletion of hRpn13^{Pru}. While the recovery process may be of interest, it is technically not feasible for this cell type. RPMI 8226 cells are suspension cells that grow very slowly. Cultures are maintained by keeping 1/2 or 1/3 of the “old” conditioned medium. 0.2-0.3 million cells/mL in T75 flask take 2 days to reach 40-50 % confluency, which is used for treatment, and following 24 hours treatment with 20 μ M XL44, more than half of cells are not viable (as we show apoptosis is triggered in these hRpn13^{Pru} containing cells). Without conditioned medium and at low density the cells take at least three days to recover and a week to divide, making it very hard to determine the timeframe within hRpn13^{Pru} is restored following discontinuation of XL44. We respectfully disagree that this data is crucial for supporting our findings and publication of the manuscript. As mentioned above, we acquired a $d_{1/2}$ value (differential protein abundance following treatment of RPMI 8226 cells with 20 μ M XL44) and found half depletion of hRpn13^{Pru} by XL44 took \sim 12 hours (Fig. 5d).